# TRUTH IN THE FEW: HIGH-VALUE DATA SELECTION FOR EFFICIENT MULTI-MODAL REASONING

## ABSTRACT

While multi-modal large language models (MLLMs) have made significant progress in complex reasoning tasks via reinforcement learning in the post-training phase, it is commonly believed that extensive training data is necessary for improving multi-modal reasoning ability, inevitably leading to data redundancy and substantial computational costs. However, *can smaller high-value datasets match or outperform full corpora for multi-modal reasoning in MLLMs?* In this work, we challenge this assumption through a key observation: meaningful multi-modal reasoning during post-training is triggered by only a sparse subset of training samples, termed *cognitive samples*, whereas the majority contribute marginally. Building on this insight, we propose a novel data selection paradigm termed ***R**easoning **A**ctivation **P**otential (RAP)*, which identifies cognitive samples by estimating each sample's potential to stimulate genuine multi-modal reasoning by two complementary estimators: 1) *Causal Discrepancy Estimator (CDE)* based on the potential outcome model principle, eliminates samples that overly rely on language priors by comparing outputs between multi-modal and text-only inputs; 2) *Attention Confidence Estimator (ACE)*, which exploits token-level self-attention to discard samples dominated by irrelevant but over-emphasized tokens in intermediate reasoning stages. Moreover, we introduce a *Difficulty-aware Replacement Module (DRM)* to substitute trivial instances with cognitively challenging ones, thereby ensuring complexity for robust multi-modal reasoning. Experiments on six datasets show that our RAP method consistently achieves superior performance *using only 9.3% of the training data, while reducing computational costs by over 43%*. Our code is available at `https://anonymous.4open.science/r/RAP-39FF`.

## 1 INTRODUCTION

Improving complex reasoning in multi-modal large language models (MLLMs) Qwen et al. (2025); OpenAI (2023) remains a fundamental challenge. While large-scale reinforcement learning (RL) during post-training Guo et al. (2025); Yang et al. (2025) has shown promise in enhancing reasoning capability, the prevailing assumption OpenAI (2024); Team et al. (2025)

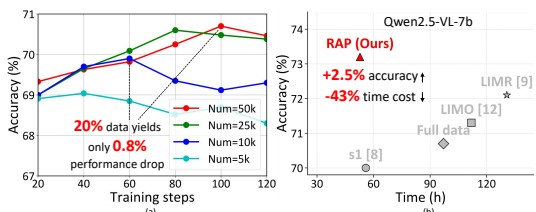

Figure 1: Comparison of (a) accuracy under varying training dataset sizes and (b) performance–efficiency trade-offs on various methods.

suggests that scaling training data is a necessary condition for developing advanced reasoning ability, thus leading to data redundancy and substantial training costs. Recent studies Muennighoff et al. (2025); Li et al. (2025) indicate that LLMs trained on high-quality curated datasets can outperform those trained on full corpora. However, it remains unclear whether this principle generalizes to multi-modal contexts, where effective cross-modal integration is important. This raises a critical question: *can smaller high-value data achieve competitive or superior multi-modal reasoning compared to training on full post-training corpora in MLLMs?* To investigate this, as shown in Figure 1(a), we empirically analyze the effect of data scale on multi-modal reasoning performance. Notably, *training with only 20% of the data leads to merely a 0.8% performance degradation compared to the full dataset*, suggesting that indiscriminate data scaling may have minimal or even negative effects. We hypothesize that such data augmentation

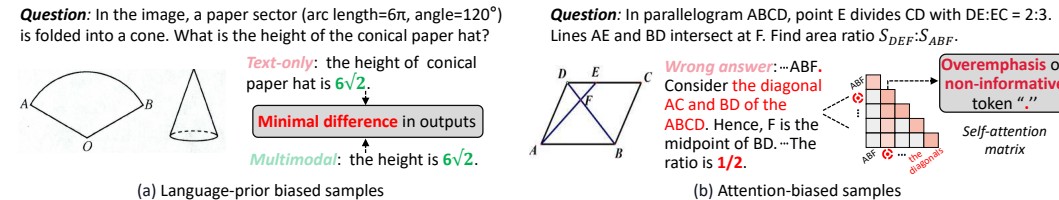

**Question**: In the image, a paper sector (arc length=6π, angle=120°) is folded into a cone. What is the height of the conical paper hat?

*Text-only*: the height of conical paper hat is **6√2**.

**Minimal difference** in outputs

*Multimodal*: the height is **6√2**.

(a) Language-prior biased samples

**Question**: In parallelogram ABCD, point E divides CD with DE:EC = 2:3. Lines AE and BD intersect at F. Find area ratio $S_{DEF}$:$S_{ABF}$.

*Wrong answer*: …ABF. Consider the diagonal AC and BD of the ABCD. Hence, F is the midpoint of BD. …The ratio is **1/2**.

**Overemphasis** on **non-informative** token ".".

*Self-attention matrix*

(b) Attention-biased samples

Figure 2: Illustrative examples for two ineffective training sample types: (a) language-prior biased samples and (b) attention-biased samples.

diminishes the influence of high-value samples, termed *cognitive samples*, which are essential for guiding effective cross-modal integration during reasoning.

To validate this assumption, we analyze the characteristics of training samples and find that most fail to encourage joint attention to both modalities during reasoning. Specifically, we identify two main types of ineffective samples: 1) *Language-prior biased samples* (Figure 2(a)), where the model produces nearly indistinguishable outputs given text-only and multi-modal inputs due to over-reliance on language priors Han et al. (2022); Leng et al. (2024). Such samples enable models to solve tasks with minimal utilization of visual semantics, thus impairing their ability to cross-modal integration. (2) *Attention-biased samples* (Figure 2(b)), where the model over-attends to semantically irrelevant tokens (*e.g.*, the punctuation "."), thereby obstructing the exploration of crucial cross-modal relationships. The above findings highlight the need to prioritize cross-modal interactions in data selection for multi-modal reasoning. However, existing data selection methods rely on unimodal textual quality, such as human-annotated difficulty estimation Li et al. (2025) or reward-based sampling Ye et al. (2025). These approaches not only incur substantial manual annotation costs Muennighoff et al. (2025); Ye et al. (2025) and considerable filtering time Li et al. (2025) in the post-training phase, but also fail to estimate whether samples effectively facilitate cross-modal integration.

Motivated by the above observations, we propose a novel data selection paradigm termed **R**easoning **A**ctivation **P**otential (RAP) for enhancing multi-modal reasoning while reducing training costs. RAP aims to identify cognitive samples that effectively trigger multi-modal reasoning during RL post-training. Specifically, RAP estimates the reasoning potential of each sample through two complementary perspectives: *output-level reasoning discrepancy* and *process-level reasoning confidence*. For the former, we are inspired by the intuition that if model predictions remain invariant regardless of visual input presence, the model may merely exploit linguistic biases rather than engage in genuine multi-modal reasoning. We formalize this notion through *Causal Discrepancy Estimator (CDE)*, which employs the Potential Outcome Model (POM) to estimate the causal effect of input modality on model predictions by simulating counterfactual outcomes, *i.e.*, what the model would output if one modality were removed. Consequently, CDE effectively eliminates *language-prior biased samples* by measuring discrepancies between multi-modal and text-only predictions.

However, relying solely on output-level measures neglects the reliability of internal reasoning dynamics. Therefore, we propose the *Attention Confidence Estimator (ACE)* to model the quality of internal reasoning behavior based on token-level attention distributions, thus removing *attention-biased samples* characterized by high attention to irrelevant tokens. Despite their efficacy, combining these two estimators alone might retain overly simplistic samples while discarding challenging yet valuable instances, thereby constraining the model's reasoning upper bound. To address this limitation, we propose a *Difficulty-aware Replacement Module (DRM)* to replace trivial samples with suitable challenging alternatives, which ensure sufficient data complexity for robust multi-modal reasoning. Finally, the results in Figure 1(b) demonstrate that RAP achieves state-of-the-art performance with only 5,159 samples, compared to the full dataset of 54,931 samples, while reducing training costs by over 43%. These findings validate our insight that data quality is more important than blind data scaling for multi-modal reasoning in RL, revealing the "*truth in the few*" phenomenon.

**Our main contribution:** 1) We reveal a "*truth in the few*" phenomenon that smaller high-quality datasets can outperform full corpora for multi-modal reasoning in MLLMs. 2) We propose two novel estimators: a *Causal Discrepancy Estimator (CDE)* to eliminate samples that overly rely on language priors, and an *Attention Confidence Estimator (ACE)* to filter out attention-biased samples with irrelevant semantic focus. 3) We introduce a *Difficulty-aware Replacement Module (DRM)* to preserve sufficient data complexity, effectively improving the model's reasoning performance ceiling.

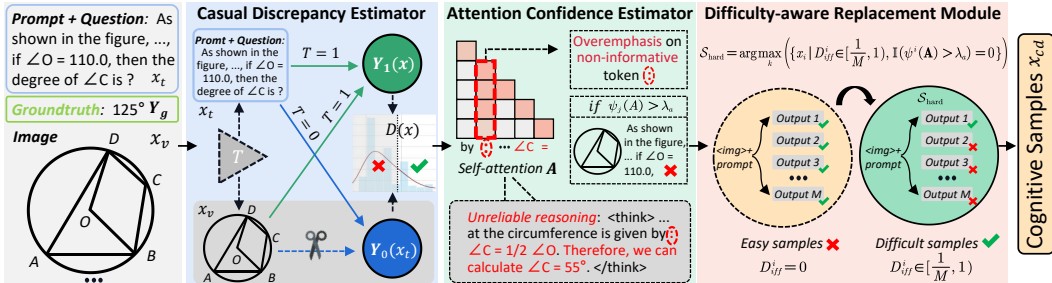

Figure 3: The overall pipeline of our RAP method. First, the Causal Discrepancy Estimator (CDE) filters out samples that overly rely on language priors. Then, the Attention Confidence Estimator (ACE) excludes attention-biased samples. Finally, the Difficulty-aware Replacement Module (DRM) replaces trivial instances with challenging ones, yielding a refined subset of cognitive samples.

## 2 RELATED WORK

**Reinforcement Learning for (M)LLM Reasoning.** Reinforcement learning (RL) has become a key paradigm for improving the reasoning ability of both LLMs and MLLMs. Recent methods Lai et al. (2024); Chowdhury et al. (2024); Rafailov et al. (2023) extend RL beyond human preference alignment, explicitly achieving reasoning improvement via policy-gradient algorithms such as Proximal Policy Optimization (PPO) Schulman et al. (2017) and reward-centric optimizations including RLOO Ahmadian et al. (2024) and GRPO Guo et al. (2025). Moreover, several methods Peng et al. (2025); Yang et al. (2025); Liu et al. (2025); Wang et al. (2024) explore the RL to enhance the visual reasoning in MLLMs. However, these methods typically rely on large-scale data Yao et al. (2024), which fails to consider the quality of training samples. Hence, we introduce a novel RAP method to select valuable samples, ensuring that the training process stimulates effective multi-modal reasoning.

**Data Selection for Reasoning.** Traditional methods Guo et al. (2025); Team et al. (2025); Yao et al. (2024) generally highlight the importance of data scaling, suggesting that larger data volumes can lead to better performance. Contrary to this, recent methods Ye et al. (2025); Li et al. (2025) demonstrate that curated datasets outperform those trained on full corpora in textual reasoning tasks. For example, LIMO Ye et al. (2025) and s1 Muennighoff et al. (2025) demonstrate that models trained with curated samples perform better compared to models trained on larger datasets. Inspired by these, we focus on whether and how a minimal but valuable dataset can enhance reasoning within multi-modal contexts.

## 3 METHOD

**Overview of RAP Method.** As shown in Figure 3, our RAP method aims to identify high-value training samples $x_{cd}$, termed cognitive samples, that effectively stimulate multi-modal reasoning in MLLMs during RL post-training. Given a training instance $x = (x_t, x_v)$, we estimate its reasoning activation potential from two perspectives. First, we adopt the potential outcome model to quantify the output-level discrepancy $D$ between the model predictions under multi-modal inputs, $Y_1(x)$, and text-only inputs, $Y_0(x_t)$. Samples with low discrepancy values be considered as language-prior biased and are discarded accordingly. Second, we compute a confidence score $\psi(A)$ from the self-attention matrix $A \in \mathbb{R}^{d \times d}$ of the model's final layer to assess the model's focus on meaningful tokens. Samples with attention focused on irrelevant tokens, below a threshold $\lambda_a$, are excluded as attention-biased. Moreover, we replace overly easy samples with an equal number of hard examples potentially missed by the initial model due to limited reasoning capacity, thereby enhancing the model's reasoning upper bound. The resulting cognitive samples form a refined training dataset that supports more efficient and robust multi-modal reasoning in MLLMs. Finally, we utilize these cognitive samples $x_{cd}$ to optimize the model by maximizing the objective of GRPO Guo et al. (2025).

### 3.1 CAUSAL DISCREPANCY ESTIMATOR

To identify samples where the model genuinely engages in multi-modal reasoning, rather than overly relying on language priors, we interpret modality influence in reasoning as output discrepancy between multi-modal and text-only inputs, formulated under the Potential Outcome Model (POM).

**Background of Potential Outcome Model.** The foundation of causal inference is rooted in the Neyman-Rubin POM Rubin (1974; 2005), which aims to estimate the effect of a treatment $T$ on an outcome $\boldsymbol{Y}$ for individuals described by covariates $\boldsymbol{X}$. In this work, we consider the treatment variable $T$ to be binary, *i.e.*, $T \in \{0, 1\}$. Under this framework, each unit $u$ is associated with two potential outcomes: $\boldsymbol{Y}_1(u)$, the outcome if the unit receives the treatment ($T = 1$), and $\boldsymbol{Y}_0(u)$, the outcome under control ($T = 0$). The Individual Treatment Effect (ITE) Johansson et al. (2016); Shalit et al. (2017) is defined as the difference $\boldsymbol{Y}_1(u) - \boldsymbol{Y}_0(u)$. However, due to the fundamental problem of causal inference Pearl (2009); Hammerton & Munafò (2021), only one of these outcomes can ever be observed for a given individual, rendering the ITE fundamentally unidentifiable. To address this, the prior work Abrevaya et al. (2015) proposes the Conditional Average Treatment Effect (CATE), which represents the expected treatment effect conditioned on covariates:

$$\mathbb{E}[\boldsymbol{Y}_1 - \boldsymbol{Y}_0 \mid \boldsymbol{X} = x] = \mathbb{E}[\boldsymbol{Y} \mid T = 1, \boldsymbol{X} = x] - \mathbb{E}[\boldsymbol{Y} \mid T = 0, \boldsymbol{X} = x], \tag{1}$$

where $x$ are the observed covariates of the unit. More theoretical details are provided in Appendix.A.1.

**Output-level Discrepancy Calculation.** Inspired by the intuition that a model generating nearly identical outputs in the presence or absence of visual input may fail to use multi-modal information for reasoning, we employ the POM to formalize the influence of the visual modality on model predictions, by defining outcomes under distinct treatment conditions. Specifically, we treat the presence of the image as a binary treatment variable $T \in \{0, 1\}$, where $T = 1$ indicates the inclusion of the image and $T = 0$ denotes its absence. Given an input $x = (x_t, x_v)$, we define two potential outcomes: $\boldsymbol{Y}_1(x)$, the model's output given both text and image, and $\boldsymbol{Y}_0(x_t)$, the counterfactual output when only the text is provided. The text-only output $\boldsymbol{Y}_0(x_t)$ can be calculated as follows:

$$y_i \in \boldsymbol{Y}_0(x_t) \sim \text{softmax}\left[\log_\theta\left(y_i \mid x_t\right)\right], \tag{2}$$

The multi-modal output $\boldsymbol{Y}_1(x)$ can be calculated as follows:

$$y_i \in \boldsymbol{Y}_1(x) \sim \text{softmax}\left[\log_\theta\left(y_i \mid x_v, x_t\right)\right], \tag{3}$$

To quantify the discrepancy between model outputs $\boldsymbol{Y}_0(x_t)$ and $\boldsymbol{Y}_1(x)$ under multi-modal and text-only inputs, we compute the consistency of these model outputs with the ground truth $\boldsymbol{Y}_g$. If the model's output matches the ground truth in a given condition, we assign a value of 1; otherwise, a value of 0. The discrepancy $D(x)$ for each sample $x$ is then quantified as the normalized difference in the number of correct predictions between these conditions, which can be formulated as:

$$D(x) = \mathbb{E}\left[\mathbb{I}(\boldsymbol{Y}_1 = \boldsymbol{Y}_g) - \mathbb{I}(\boldsymbol{Y}_0 = \boldsymbol{Y}_g) \mid x\right] = \frac{1}{M} \sum_{i=1}^{M} \left[\mathbb{I}(Y_1(x^{(i)}) = Y_g^{(i)}) - \mathbb{I}(Y_0(x_t^{(i)}) = Y_g^{(i)})\right], \tag{4}$$

where $M$ is the number of rollout outputs generated for each sample set in GRPO, and $\mathbb{I}(\cdot)$ is the indicator function that equals 1 if the condition is true, and 0 otherwise. Based on the mean and standard deviation of discrepancies across all samples, we set a threshold for the discrepancy score. Specifically, samples with discrepancy less than $\mu_c + \lambda_c \cdot \sigma_c$, where $\mu_c$ is the mean discrepancy, $\sigma_c$ is the standard deviation, and $\lambda_c$ is a tunable hyperparameter, are excluded from the training set.

### 3.2 ATTENTION CONFIDENCE ESTIMATOR

While the CDE identifies samples requiring multi-modal reasoning from an output-level perspective, it does not assess the quality of internal reasoning processes. Recent studies Huang et al. (2024); Wang et al. (2023) reveal an insightful phenomenon: tokens with excessive attention weights can dominate the prediction without using meaningful semantics. Motivated by this, we explicitly quantify the internal reasoning quality via self-attention distributions, thus filtering out *attention-biased samples*.

**Attention Confidence Formulation.** Given an input $x = (x_t, x_v)$ to a transformer-based MLLM, we denote the self-attention matrix $\boldsymbol{A} \in \mathbb{R}^{d \times d}$ from its final transformer layer Rohekar et al. (2023) as:

$$A_{i,j} = \text{softmax}\left(\frac{\boldsymbol{Q}_i \boldsymbol{K}_j^\top}{\sqrt{d}}\right), \tag{5}$$

where $\boldsymbol{Q}_i, \boldsymbol{K}_j \in \mathbb{R}^d$ represent the query and key vectors for token positions $i$ and $j$.

To systematically characterize attention-bias patterns, ACE analyzes the entire self-attention matrix $\boldsymbol{A}$. An attention-biased pattern at token position $j$ is identified if the corresponding attention column exhibits a pronounced concentration of attention weights, identifying excessive reliance on a single token. Formally, the degree of attention bias at position $j$, $\psi_j(\boldsymbol{A})$ is quantified by computing a multiplicative attention score across subsequent token interactions:

$$\psi_j(\boldsymbol{A}) = \prod_{i=j}^{L}(\sigma \cdot A_{i,j}), \tag{6}$$

where $\sigma$ is a scaling factor ensuring numerical stability and emphasizing prominent attention patterns. $L$ denotes the total length of the input sequence. The $\psi_j(A)$ metric effectively quantifies attention confidence, with elevated values indicating unreliable multi-modal reasoning. We define a position $j$ as attention-biased if $\psi_j(A)$ exceeds a threshold $\lambda_a$. Instances containing more than one attention-biased position are filtered out as attention-biased samples.

### 3.3 Difficulty-aware Replacement Module

Although CDE and ACE select valuable samples, they inevitably limit the reasoning upper bound due to the exclusion of challenging yet informative samples. For example, in the CDE selection process, if the output discrepancy threshold is set above 0.2, a scenario may arise where a text-only model consistently produces incorrect outputs across all five trials, whereas multi-modal outputs succeed once in five trials (more details are provided in Appendix A.2.3). Such challenging yet valuable samples, which could significantly contribute to improving reasoning, are thus discarded. This exclusionary process reduces the complexity of the training data, thereby constraining the model to achieve more complex reasoning ability. To address this, we introduce a Difficulty-aware Replacement Module (DRM) to refine the selected sample. First, we define the difficulty score $D_{iff}^i$ to quantify the challenge of correctly answering a sample, which can be defined as:

$$D_{iff}^i = 1 - \frac{\sum_{j=1}^{M} c_{i,j}}{M}, \tag{7}$$

where $c_{i,j}$ denotes the correctness of the $j$-th rollout generation for the $i$-th sample, and $M$ is the total number of outputs in the group. A higher $D_{iff}^i$ indicates greater difficulty. In particular, the DRM involves two steps: First, we exclude easy samples, denoted as $D_{iff}^i = 0$, which are characterized by consistent correct answers across all trials. Second, we reintroduce hard samples that have been previously discarded due to difficulty but are still valuable for training. Specifically, based on the difficulty score $D_{iff}^i$, the set of reintroduced samples $\mathcal{S}_{\text{hard}}$ is given by:

$$\mathcal{S}_{\text{hard}} = \mathrm{argmax}_k \left( \{x_i \mid D_{iff}^i \in [\frac{1}{M}, 1), \mathbb{I}(\psi^i(\boldsymbol{A}) > \lambda_a) = 0\} \right), \tag{8}$$

where $k$ is the number of easy samples. This formulation identifies the top-$k$ most challenging samples according to the difficulty metric $D_{iff}^i$, while excluding those that do not meet the specified criteria. Note that the hardest samples, $i.e.$, $D_i = 1$, would be neglected, as they are demonstrated to be meaningless for training Huang et al. (2025). This DRM can enhance the upper bound of the model's ability to handle complex tasks, without introducing data redundancy and training costs.

Finally, by filtering through CDE and ACE and refining with DRM, we ensure that the model is trained with cognitive samples $x_{cd}$, thereby enhancing the multi-modal reasoning ability, while simultaneously reducing training costs and data redundancy.

## 4 Experiments

**Training dataset.** Main results in the Table 1 are based on models trained with the *MM-Eureka* dataset Meng et al. (2025), a high-quality multi-modal dataset for mathematical reasoning. To further validate the generalization of our RAP method, we evaluate models trained on the subset of *Mulberry-260k* dataset Yao et al. (2024), a multi-modal learning-to-reason-and-reflect dataset.

**Evaluation.** Similar to Yao et al. (2024); Meng et al. (2025), we evaluate models on both mathematical and general multi-modal reasoning tasks using the $pass@1$ metric, where $pass@1$ measures

Table 1: Comparison with state-of-the-art methods. Experiments are conducted using the Qwen2.5-VL-3b Qwen et al. (2025) and Qwen2.5-VL-7b Qwen et al. (2025), employing GRPO as the RL method. "Time" denotes the total computation cost, including data selection and training. "*" represents the relaxing certain selection criteria for s1 and LIMO. Bold font denotes the best result.

| Method | Sample | Time (h) ↓ | MathVista ↑ | MMStar ↑ | MathVerse ↑ | WeMath ↑ | MMVet ↑ | LogicVista ↑ | Avg. ↑ |
|---|---|---|---|---|---|---|---|---|---|
| Qwen2.5-VL-7b | - | - | 68.70 | 56.07 | 39.31 | 35.90 | 59.13 | 44.52 | 50.61 |
| Qwen2.5-VL-7b-Full | 54,931 | 93.2 | 70.70 | 61.53 | 48.43 | 38.67 | 60.51 | 46.09 | 54.32 |
| Qwen2.5-VL-7b-s1 (2025) | 1,000 | 55.9 | 68.50 | 61.80 | 45.79 | 35.05 | 61.05 | 45.86 | 53.01 |
| Qwen2.5-VL-7b-s1* (2025) | 6,109 | 64.6 | 68.60 | 61.46 | 45.34 | 34.82 | 60.23 | 45.29 | 52.62 |
| Qwen2.5-VL-7b-LIMO (2025) | 4,093 | 111.9 | 69.90 | 61.33 | 45.74 | 34.67 | 59.08 | 44.74 | 52.58 |
| Qwen2.5-VL-7b-LIMO* (2025) | 5,847 | 120.6 | 69.20 | 61.27 | 46.03 | 34.51 | 58.22 | 44.53 | 52.29 |
| Qwen2.5-VL-7b-LIMR (2025) | 8,136 | 122.0 | 71.10 | 62.12 | 48.02 | 41.21 | 62.86 | 45.81 | 55.19 |
| **Qwen2.5-VL-7b-RAP (Ours)** | 5,159 | **52.8** | **73.20** | **62.53** | **48.65** | **42.00** | **63.31** | **46.53** | **56.04** |
| Qwen2.5-VL-3b | - | - | 61.30 | 54.46 | 9.01 | 21.62 | 51.81 | 39.59 | 39.63 |
| Qwen2.5-VL-3b-Full | 54,931 | 46.5 | 64.50 | 55.25 | 38.35 | 28.29 | 53.41 | 40.03 | 46.64 |
| Qwen2.5-VL-3b-s1 (2025) | 1,000 | 38.4 | 62.60 | 54.53 | 37.41 | 27.52 | 53.02 | 39.59 | 45.78 |
| Qwen2.5-VL-3b-s1* (2025) | 5,103 | 42.7 | 62.30 | 54.18 | 35.89 | 23.43 | 52.93 | 39.82 | 44.76 |
| Qwen2.5-VL-3b-LIMO (2025) | 2,679 | 94.4 | 61.80 | 54.73 | 35.33 | 24.48 | 53.12 | 39.14 | 44.77 |
| Qwen2.5-VL-3b-LIMO* (2025) | 4,808 | 98.8 | 62.40 | 54.66 | 35.26 | 24.81 | 52.03 | 39.03 | 44.69 |
| Qwen2.5-VL-3b-LIMR (2025) | 21,303 | 60.9 | 63.10 | 54.66 | 35.43 | 26.76 | 53.25 | 40.95 | 45.69 |
| **Qwen2.5-VL-3b-RAP (Ours)** | 4,374 | **32.0** | **64.90** | **55.67** | **39.34** | **29.33** | **54.63** | **41.61** | **47.58** |

the percentage of problems correctly solved on the first attempt, under a zero-shot setting. For mathematical reasoning, we assess the model's ability on four benchmarks: MathVista Lu et al. (2024), MMStar Chen et al. (2024a), MathVerse Zhang et al. (2024), and WeMath Qiao et al. (2024). For universal reasoning, we evaluate on MMVet Yu et al. (2024) and LogicVista Xiao et al. (2024).

**Implementation details.** Following prior methods Meng et al. (2025); Yao et al. (2024), we conduct our primary experiments on mathematical reasoning tasks, employing Qwen2.5-VL-3B and Qwen2.5-VL-7B Qwen et al. (2025) as baseline models. First, we apply RAP to select cognitive samples using the initial model without any training. These samples are then used to train models within the EasyR1 Sheng et al. (2024), employing the AdamW with a learning rate of 1e-6. Full-data training requires 1 epoch (107 steps), all others use 40 training steps with a batch size of 512 across 8 GPUs. For accelerated generation in GRPO Guo et al. (2025), we utilize the vLLM package Kwon et al. (2023). Finally, we set the $\sigma$ to 2.0, $\lambda_c$ and $\lambda_a$ to 0.5 and 0.1 for the CDE and ACE, respectively.

## 4.1 OVERALL COMPARISON RESULTS

**Comparing methods.** We compare our approach with existing data selection methods, including: 1) *s1* Muennighoff et al. (2025), which utilizes the large-scale MLLMs to identify high-quality data; 2) *LIMO* Ye et al. (2025) that designs a difficulty-aware selection method to identify crucial samples; and 3) *LIMR* Li et al. (2025), which employs learning impact measurement to select a subset of training samples. Moreover, we also evaluate models trained on the full dataset (Full) as the baseline.

**Comparisons with state-of-the-art methods.** The results shown in Table 1 reveal several key findings: 1) RAP consistently outperforms models trained with full corpora on all datasets. Remarkably, these improvements are achieved using only 9.5% or 7.9% of training data while reducing training time by 43% or 31%, supporting our hypothesis "*truth in the few*" that selected cognitive samples can achieve more effective multi-modal reasoning. 2) Moreover, the RAP shows a significant improvement of 7.33% and 6.95% over the LIMO and s1 on WeMath, which overly rely on manual selection. For fairness, we strictly followed the data selection from original s1 and LIMO. Moreover, to avoid misunderstanding about performance drops from smaller datasets, we also provide comparisons with a more relaxed data selection strategy (details in Appendix A.3.1) for these, where RAP consistently outperforms both methods, confirming the effectiveness of focusing on the potential of each sample.

**Effectiveness of RAP on different base models.** As shown in Table 2, our method consistently surpasses other recent data selection methods when applied to the base model InternVL3-2b Chen et al. (2024b). This outcome highlights the broad applicability and generalizability of the RAP framework, as the introduced CDE and ACE components effectively select training samples that activate the model's multi-modal reasoning capability. Crucially, these components do not rely on exploiting model-specific inductive biases, thereby ensuring RAP's adaptability on a wide range of model architectures. More results are available in Appendix A.4.3.

**Effectiveness of RAP for various RL methods and training datasets.** To further validate the generalizability of RAP, Table 3 presents results using the Qwen2.5-VL-7B model, trained under

Table 2: Comparison with state-of-the-art methods using Qwen2.5-VL-7b Qwen et al. (2025) with the RL method RLOO, evaluating RAP on different training datasets and RL algorithms.

| Method | MathVista | MMVet | We-Math | Avg. |
|---|---|---|---|---|
| Qwen2.5-VL-7b | 68.70 | 59.13 | 35.90 | 54.58 |
| Qwen2.5-VL-7b-Full | 69.10 | 60.32 | 36.95 | 55.46 |
| Qwen2.5-VL-7b-s1 (2025) | 68.50 | 59.96 | 35.06 | 54.51 |
| Qwen2.5-VL-7b-LIMO (2025) | 68.80 | 60.11 | 35.23 | 54.71 |
| Qwen2.5-VL-7b-LIMR (2025) | 68.90 | 60.71 | 36.74 | 55.45 |
| **Qwen2.5-VL-7b-RAP (Ours)** | **69.20** | **61.33** | **37.05** | **55.86** |

Table 3: Comparison with state-of-the-art methods using InternVL3-2b Chen et al. (2024b) with the RL method GRPO, evaluating our RAP method across different model architectures.

| Method | MathVista | MMVet | We-Math | Avg. |
|---|---|---|---|---|
| InternVL3-2b | 56.10 | 58.22 | 12.06 | 42.13 |
| InternVL3-2b-Full | 57.20 | 59.86 | 12.84 | 43.30 |
| InternVL3-2b-s1 (2025) | 56.80 | 60.47 | 11.63 | 42.97 |
| InternVL3-2b-LIMO (2025) | 56.70 | 59.82 | 11.92 | 42.81 |
| InternVL3-2b-LIMR (2025) | 57.10 | 61.33 | 12.44 | 43.62 |
| **InternVL3-2b-RAP (Ours)** | **57.40** | **62.02** | **13.05** | **44.16** |

two different configurations: 1) the RLOO RL paradigm Ahmadian et al. (2024), and 2) the reduced Mulberry-10K dataset, a subset of Mulberry-260K Yao et al. (2024). Despite these variations, RAP maintains consistent superiority compared to other methods, suggesting its generalization to different RL strategies and training datasets. We attribute this robustness to cognitive samples that facilitate genuine multi-modal reasoning rather than simply fitting data distributions.

## 4.2 FURTHER ANALYSIS

**Ablation study.** As presented in Table 4, we list the following conclusions: 1) The comparison between No.0 and No.3 indicates that integrating CDE and ACE can improve multi-modal reasoning in MLLMs. These results underscore the efficacy of RAP in eliminating language-prior biased and attention-biased samples, thereby enabling models to focus on essential "*cognitive samples*".

2) Comparing No.1 with No.3 shows that only using the CDE can improve performance, but worse than the full RAP, which identifies critical samples by detecting output discrepancies, overlooking the intermediate reasoning. 3) Moreover, the comparison between No.2 and No.3 demonstrates that the DRM can further refine the reasoning performance by replacing easy samples with more appropriate hard samples. Such improvements demonstrate that the DRM addresses a limitation of the CDE and ACE,

Table 4: Ablation study of RAP using Qwen2.5-VL-7B.

| No. | CDE | ACE | DRM | MathVista | MMStar | MathVerse | MMVet |
|---|---|---|---|---|---|---|---|
| 0 | - | - | - | 69.10 | 59.32 | 46.07 | 58.91 |
| 1 | ✔ | | | 70.80 | 60.64 | 47.13 | 60.92 |
| 2 | | ✔ | | 70.20 | 60.21 | 46.86 | 60.68 |
| 3 | ✔ | | ✔ | 72.00 | 61.28 | 47.82 | 61.74 |
| 4 | | ✔ | ✔ | 71.50 | 60.93 | 47.90 | 61.33 |
| 5 | ✔ | ✔ | | 72.60 | 61.76 | **48.74** | 62.78 |
| 6 | ✔ | ✔ | ✔ | **73.20** | **62.53** | 48.65 | **63.31** |

*i.e.*, the tendency to retain simpler instances while neglecting challenging yet informative samples.

**Comparison on cross-modal reasoning utilization.** We evaluate the effectiveness of cognitive samples in enhancing cross-modal reasoning utilization on multi-modal reasoning tasks. We define cross-modal reasoning utilization as the proportion of instances in the MMStar dataset where the model correctly uses multi-modal inputs to answer questions, but fails when relying solely on textual inputs. As shown in Figure 5(c), models trained with cognitive samples show superior integration of cross-modal information, compared to baseline and the latest method LIMR Li et al. (2025). Such results highlight that the proposed CDE can effectively reduce models' reliance on superficial linguistic prior by filtering out training samples exhibiting excessive language bias. Hence, models are encouraged to discover the relationship between image and text, thus improving the cross-modal reasoning utilization.

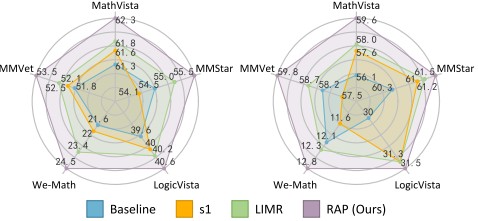

Figure 4: Cross-model generalization of cognitive samples selected by RAP. Performance with InternVL3-2B trained on samples from Qwen2.5-VL-3B (left), and vice versa (right).

**Comparison on cross-model generalization.** To examine the cross-model generalizability of cognitive samples identified by RAP, we evaluate whether cognitive samples obtained using the Qwen2.5-VL-3b are useful for improving the reasoning of a distinctly structured model, InternVL3-2b, and vice versa. Results presented in Figure 4 demonstrate that cognitive samples selected by our RAP method outperform the latest method LIMR Li et al. (2025), confirming the generalization of RAP in enhancing multi-modal reasoning for varying model frameworks.

**Analysis on hyperparameter sensitivity.** We further investigate the sensitivity of the hyperparameters $\lambda_c$ and $\lambda_a$ employed within the CDE and ACE, respectively. As shown in Figure 5(a), we visualize the distribution of output discrepancies between multi-modal and text-only inputs, revealing

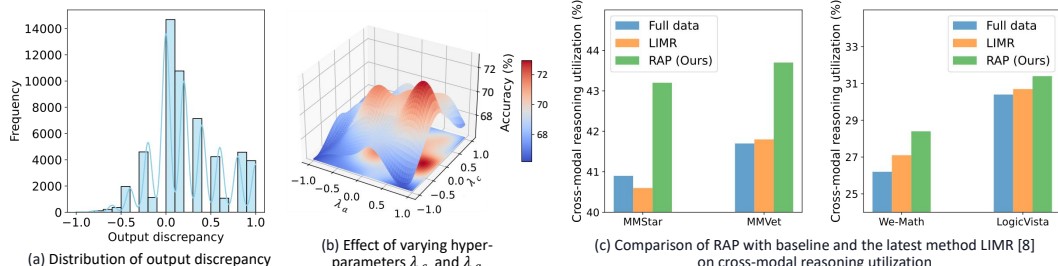

Figure 5: (a) Visualization of output discrepancies between multi-modal and text-only inputs on the full MM-Eureka training dataset. (b) Performance variation with respect to the hyperparameters $\lambda_a$ and $\lambda_c$ on MMstar. (c) Comparative analysis of multi-modal reasoning utilization on four datasets.

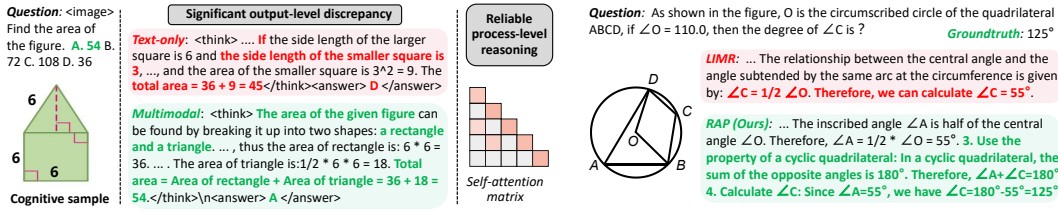

Figure 6: Illustration of (a) characteristics of cognitive samples selected by our RAP and (b) Comparison of the reasoning processes between our RAP and the state-of-the-art LIMR Li et al. (2025).

a significant presence of samples where multi-modal reasoning is not necessary to solve the task. Experimental results depicted in Figure 5(b) suggest that optimal performance is achieved with $\lambda_c = 0.1$ and $\lambda_a = 0.5$. Performance degradation is observed when both parameters fall below these optimal thresholds, with values lower than 0.1 for $\lambda_c$ and 0.5 for $\lambda_a$ leading to significant deterioration. This decline is attributed to the inherently uneven distribution of the output-level discrepancy and attention confidence, which are concentrated at the extremes.

**Qualitative analysis.** We provide a qualitative analysis by visualizing cognitive samples and comparing the reasoning processes of our RAP with the latest LIMR method as follows: 1) *Visualization of cognitive samples.* As shown in Figure 6(a), we present a typical example of the cognitive samples selected by RAP. This case suggests that cognitive samples selected by RAP exhibit two important characteristics: (a) the necessity of multi-modal information, as evidenced by the significant discrepancy between reasoning outcomes using multi-modal and text-only inputs; and (b) the avoidance of overemphasis on irrelevant or meaningless tokens, ensuring that the model focuses on informative features for accurate reasoning. 2) *Comparison case analysis.* As shown in Figure 6(b), we compare the reasoning process of our RAP method with that of the state-of-the-art LIMR Li et al. (2025). For example, the LIMR fails to integrate cross-modal information, leading to incorrect computations of both central angle and inscribed angle. In contrast, the model trained using samples selected by RAP correctly applies geometric principles and multi-modal integration to arrive at the correct solution. These comparisons show the advantage of training with cognitive samples derived through our RAP, which enables the model to effectively leverage multi-modal information.

## 4.3 KEY INSIGHTS AND DISCUSSION

**Effect of RAP on reasoning upper bound.** As illustrated in Figure 7(a), our RAP method converges faster than the baseline, achieving optimal performance within 40 training steps compared to 100 for the baseline. These results demonstrate RAP's efficiency in enhancing multi-modal reasoning while reducing training overhead and data redundancy. However, further analysis reveals differences between the full RAP model and its variant using only ACE and CDE, particularly during later training stages. This occurs because ACE and CDE inevitably retain easy examples, while discarding challenging yet informative samples. For example, multi-modal examples predicted correctly in only one out of five attempts but consistently mispredicted under text-only conditions may be mistakenly eliminated when the discrepancy threshold $\lambda_c$ exceeds 0.2. Such filtering reduces dataset complexity, limiting the model's overall reasoning upper bound. To address this, we propose the Difficulty-aware

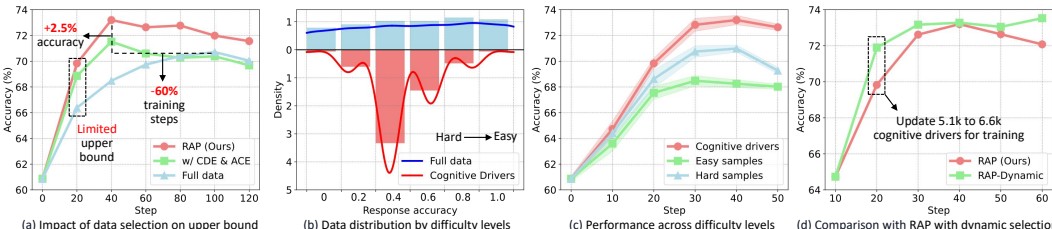

Figure 7: (a) Trade-off analysis between efficiency and performance and effect of data selection on upper bound. (b-c) Impact of varying the proportion of samples with different difficulty levels on reasoning. (c) Comparison with our RAP method and RAP augmented with dynamic selection.

Replacement Module (DRM) to explicitly replace easy samples with these informative yet challenging examples, thus elevating reasoning performance in later training stages.

**Why does less data outperform the full dataset?** The above empirical analysis shows that models trained on carefully curated data can notably outperform those trained on the entire dataset. To further elucidate the underlying mechanisms driving this "*truth in the few*" phenomenon, we examine sample difficulty distributions quantified by group-wise response accuracy in the GRPO generation paradigm in Figure 7(b). The results reveal that cognitive samples contain significantly fewer easy samples compared to the full dataset. Therefore, under conventional large-scale training, which usually restricts training to 1 or 2 epochs, the models lack sufficient repeated exposure to challenging samples, thus limiting the improvements brought from these valuable instances.

Moreover, to further validate our hypothesis, we conduct a comparison in Figure 7(c), where models are trained on equivalent numbers of easy, hard, and cognitive samples. The results reveal that the abundance of easy examples contributes little to advancing the model's reasoning capability and predominantly introduces redundant information. While challenging samples yield better performance than easy samples, they still fall short compared to cognitive samples selected by our RAP. We argue that the superiority of cognitive samples arises not merely from sample difficulty but from their ability to activate the model's multi-modal reasoning capacity. In multi-modal contexts, the training samples must facilitate the activation of the model's multi-modal reasoning. Additionally, we explore the effects of RAP-based variants in text-only scenarios, as detailed in Appendix A.3.2.

**Discussion on potential optimization during RL training.** The static nature of our current RAP method motivates us to explore potential optimizations through the dynamic adjustment of the training dataset during RL training. Specifically, we examine the evolving distribution of cognitive samples identified by RAP criteria across successive epochs. Initially, as depicted in Figure 7(d), approximately 5,000 samples meet the cognitive sample criteria. However, the number of samples that continues to satisfy this criterion progressively declines as training advances. This observed trend suggests the promise of adaptive data sampling, wherein cognitive samples are re-sampled dynamically from the entire dataset after the first and third training epochs. Preliminary experiments indicate that such adaptive strategies yield tangible improvements in model performance. However, despite these advantages, the re-screening of the entire dataset after each epoch incurs a significant computational overhead. As a result, our final approach strategically refrains from adopting this adaptive data selection in order to maintain computational efficiency. This highlights the inherent trade-off between performance enhancements and computational cost when incorporating dynamic data selection strategies into the RL training process. Further analysis is provided in Appendix A.2.2.

## 5 CONCLUSION

In this work, we introduce a Reasoning Activation Potential (RAP) data selection paradigm, which reduces training costs and improves multi-modal reasoning in MLLMs. RAP utilizes the Causal Discrepancy Estimator (CDE) and Attention Confidence Estimator (ACE) to effectively eliminate attention-biased samples and language-prior biased samples, leading to more accurate and efficient reasoning. For future work, we plan to investigate the efficacy of RAP in SFT training and introduce dynamic mechanisms to further improve the efficiency of multi-modal reasoning.

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

## A  APPENDIX

In this supplementary material, we further substantiate the contributions of our Reasoning Activation Potential (RAP) method, providing theoretical justifications, expanded empirical validations, and comprehensive implementation specifics beyond those covered in the main manuscript. Specifically, the supplementary contents are structured as follows:

- First, we present a rigorous theoretical treatment of our proposed Causal Discrepancy Estimator (CDE). Grounded in the classical Neyman-Rubin potential outcomes framework, we explicitly formalize foundational assumptions, including Stable Unit Treatment Value Assumption (SUTVA), Ignorability, and Overlap that justify causal inference within our multi-modal context.

- Second, we expand on key issues discussed in the main paper, specifically: (1) the relaxation of certain selection criteria and (2) the discussion on dynamic selection strategies.

- Furthermore, we enhance our experimental evaluation by benchmarking RAP against several challenging multi-modal reasoning datasets, and assess its performance in text-only scenarios and during the SFT phase. Additionally, we analyze the impact of different attention layer choices. Empirical results demonstrate consistent and substantial performance gains across diverse model architectures and reinforcement learning algorithms.

- We also compare RAP with state-of-the-art methods, augmenting our empirical analysis with two additional challenging evaluation datasets. This analysis includes results using different base models and RL strategies, further validating the robustness and adaptability of our approach.

- Moreover, we delineate the methodological intricacies of RAP, elaborating on the interplay among its constituent components: the Causal Discrepancy Estimator (CDE), the Attention Confidence Estimator (ACE), and the Difficulty-aware Replacement Module (DRM). We provide a detailed pseudo-code representation to transparently illustrate the computational workflow and facilitate reproducibility.

- Additionally, an in-depth comparative analysis of computational efficiency is presented, quantitatively evaluating the cost-performance trade-offs of RAP relative to existing advanced methods (*e.g.*, s1 Muennighoff et al. (2025), LIMO Ye et al. (2025), LIMR Li et al. (2025)) across varying dataset scales and model complexities.

- Finally, we include qualitative case studies and visualizations, providing concrete insights into RAP's capability to systematically identify cognitively demanding samples, thereby significantly enhancing the generalization and robustness of multi-modal reasoning models.

### A.1  POTENTIAL OUTCOMES FRAMEWORK: FOUNDATIONS AND IDENTIFICATION THEORY FOR CAUSAL DISCREPANCY ESTIMATION

#### A.1.1  PROBLEM SETTING AND NOTATION

We adopt the classical Neyman-Rubin causal model Rubin (2005) to formalize causal inference with observational data. Suppose we observe a finite sample $\{(X_i, T_i, Y_i)\}_{i=1}^n$, where for each unit $i$:

- $X_i \in \mathcal{X} \subseteq \mathbb{R}^d$ is the vector of covariates (or contexts).
- $T_i \in \{0, 1\}$ is the binary treatment indicator, where $T_i = 1$ signifies treatment and $T_i = 0$ control.
- $Y_i \in \mathbb{R}$ is the observed outcome for unit $i$.

Each unit possesses two potential outcomes $Y_i(0), Y_i(1)$, representing the outcomes we would observe under control and treatment, respectively. The observed outcome is connected to potential

outcomes via the switching equation:

$$Y_i = T_i Y_i(1) + (1 - T_i) Y_i(0), \tag{9}$$

The primary object of interest in causal inference is the Conditional Average Treatment Effect (CATE), defined formally as:

**Definition 1** (Conditional Average Treatment Effect). The CATE at a given covariate context $x$ is defined as the conditional expectation of the difference between potential outcomes:

$$D(x) := \mathbb{E}[Y(1) - Y(0) \mid X = x]. \tag{10}$$

This estimate provides personalized insights into treatment effectiveness.

### A.1.2 Key Assumptions for Identification

To rigorously identify causal quantities from observational data, the following standard assumptions are essential:

[Stable Unit Treatment Value Assumption, SUTVA] The observed outcome corresponds exactly to the potential outcome for the received treatment. Formally,

$$Y = TY(1) + (1 - T)Y(0). \tag{11}$$

This assumption implies no interference between units and well-defined treatments.
[Ignorability (Conditional Independence)] Given covariates $X$, treatment assignment is conditionally independent of potential outcomes:

$$\{Y(0), Y(1)\} T \mid X. \tag{12}$$

Ignorability ensures that there are no unobserved confounders; all variables influencing both treatment and outcomes are contained within $X$.
[Overlap (Positivity)] For every set of covariates $x$, each treatment has a strictly positive probability of being assigned:

$$0 < P(T = t \mid X = x) < 1, \quad \forall x \in \mathrm{supp}(X), t \in \{0, 1\}. \tag{13}$$

This assumption guarantees that each covariate configuration could feasibly receive either treatment.

### A.1.3 Main Identification Result

We now state the primary theoretical result that justifies the empirical estimation of CATE from observed data:

[Identification of CATE] Under Assumptions A.1.2, A.1.2, and A.1.2, the Conditional Average Treatment Effect (CATE) is identifiable from observational data as follows:

$$D(x) = \mathbb{E}[Y \mid T = 1, X = x] - \mathbb{E}[Y \mid T = 0, X = x]. \tag{14}$$

[Proof (Detailed)] We start by explicitly writing the definition of CATE:

$$D(x) = \mathbb{E}[Y(1) - Y(0) \mid X = x] = \mathbb{E}[Y(1) \mid X = x] - \mathbb{E}[Y(0) \mid X = x]. \tag{15}$$

**Step 1: Using Ignorability**  By Assumption A.1.2, potential outcomes are conditionally independent of treatment assignment given $X = x$. Thus:

$$\mathbb{E}[Y(t) \mid X = x] = \mathbb{E}[Y(t) \mid T = t, X = x], \quad t \in \{0, 1\}. \tag{16}$$

**Step 2: Consistency (SUTVA)**  Assumption A.1.2 ensures that for those units receiving treatment $t$, the potential outcome matches the observed outcome. Therefore:

$$\mathbb{E}[Y(t) \mid T = t, X = x] = \mathbb{E}[Y \mid T = t, X = x], \quad t \in \{0, 1\}. \tag{17}$$

**Step 3: Combining Steps 1 and 2**  Substituting back these relations yields:

$$D(x) = \mathbb{E}[Y \mid T = 1, X = x] - \mathbb{E}[Y \mid T = 0, X = x]. \tag{18}$$

This completes the identification proof rigorously.

### A.1.4 EMPIRICAL ESTIMATION OF CATE

Given an empirical dataset, we estimate the CATE as follows:

$$\hat{\tau}(x) = \frac{\sum_{i:T_i=1, X_i=x} Y_i}{\sum_{i:T_i=1, X_i=x} 1} - \frac{\sum_{i:T_i=0, X_i=x} Y_i}{\sum_{i:T_i=0, X_i=x} 1}. \tag{19}$$

This plug-in estimator is unbiased under our assumptions and converges in probability to $D(x)$ as sample size $n \to \infty$.

### A.1.5 THEORETICAL FOUNDATION FOR OUR PROPOSED CAUSAL DISCREPANCY ESTIMATOR

Building on the above foundations, we further articulate how the classical potential outcomes framework underpins our Causal Discrepancy Estimator (CDE). In our setting, the "treatment" variable $T$ is naturally instantiated as the presence ($T = 1$) or absence ($T = 0$) of the visual modality for a given sample. The "outcome" $Y$ is defined as the model's output, *e.g.*, correctness indicator or prediction accuracy, given both the sample and the model in its current state. Thus, our approach directly leverages the rigorous Neyman-Rubin model, enabling principled reasoning about the causal effect of modality on model behavior.

**Mapping CDE to Potential Outcomes.**  Formally, for each instance $x$, we define two potential outcomes:

$$Y(1) = \text{Model's output given both image and text}, \quad Y(0) = \text{Model's output given text only}. \tag{20}$$

Following the established causal inference tradition, the estimand of interest is the CATE, which in this context quantifies the expected causal effect of incorporating visual information at a given context $x$:

$$D(x) = \mathbb{E}[Y(1) - Y(0) \mid X = x]. \tag{21}$$

This quantity precisely captures the degree to which multi-modal signals influence model predictions—a direct operationalization of our desideratum in data selection.

**Assumptions in the Multi-Modal Setting.** Consistent with the standard causal literature, our analysis presumes three fundamental assumptions:

- *Stable Unit Treatment Value Assumption (SUTVA):* For each sample, the observed output matches the potential outcome under the received "treatment" (*i.e.*, input modality). This is a reasonable presumption given that each model response is deterministically linked to its input.
- *Ignorability (Conditional Independence):* Given the context $X$, the assignment of visual modality (*i.e.*, whether the image is present) is independent of the potential outcomes. In practice, this is realized by simulating both $T = 1$ and $T = 0$ conditions for each sample in our CDE protocol.
- *Overlap (Positivity):* For all $x$, both $P(T = 1 \mid X = x)$ and $P(T = 0 \mid X = x)$ are strictly positive, which is satisfied by design in our evaluation scheme.

**Main Identification Result and Implications.**  Under these assumptions, we have the following identification result:

$$D(x) = \mathbb{E}[Y \mid T = 1, X = x] - \mathbb{E}[Y \mid T = 0, X = x]. \tag{22}$$

The proof follows the standard two-step reduction: (1) Ignorability enables us to swap the conditioning on $T$ and $X$, and (2) SUTVA justifies replacing potential outcomes with observed outcomes for realized treatments. For completeness, we provide a detailed argument below.

[Proof] Starting from the definition of the Conditional Average Treatment Effect (CATE), we have:

$$D(x) = \mathbb{E}[Y(1) - Y(0) \mid X = x] = \mathbb{E}[Y(1) \mid X = x] - \mathbb{E}[Y(0) \mid X = x]. \tag{23}$$

To proceed, we explicitly invoke the Ignorability assumption (Conditional Independence) (Assumption A.1.2). This assumption ensures the conditional independence of the potential outcomes from the treatment assignment given the covariates $X$, formally stated as:

$$\{Y(1), Y(0)\} T \mid X. \tag{24}$$

Under this condition, for each treatment level $t \in \{0, 1\}$, we have the equality of conditional expectations:

$$\mathbb{E}[Y(t) \mid X = x] = \mathbb{E}[Y(t) \mid T = t, X = x]. \tag{25}$$

Next, we leverage the Stable Unit Treatment Value Assumption (SUTVA) (Assumption A.1.2), often termed the consistency assumption. This assumption ensures that the observed outcome under the assigned treatment precisely equals the corresponding potential outcome. Hence, the consistency condition formally yields:

$$\mathbb{E}[Y(t) \mid T = t, X = x] = \mathbb{E}[Y \mid T = t, X = x]. \tag{26}$$

Combining these two essential steps, we deduce the following identification result:

$$D(x) = \mathbb{E}[Y \mid T = 1, X = x] - \mathbb{E}[Y \mid T = 0, X = x]. \tag{27}$$

This final expression explicitly relates the causal estimand to observable conditional expectations. Consequently, this identification result rigorously supports empirical estimation procedures commonly used in practice, providing theoretical justification for deriving causal interpretations from observational data.

**Operationalization in CDE.** For each instance $x$, we construct both the multi-modal input $(x_t, x_v)$ and the text-only input $x_t$, and compute the difference in model outputs under these two conditions. The empirical analog of $D(x)$ becomes:

$$D(x) = \frac{1}{M} \sum_{i=1}^{M} \left[ I(Y_1^{(i)} = Y_g) - I(Y_0^{(i)} = Y_g) \right], \tag{28}$$

where $Y_1^{(i)}$ and $Y_0^{(i)}$ denote the model's predictions on $x$ with and without the visual modality, $Y_g$ is the ground truth, and $M$ is the number of model rollouts or generations considered. This construction directly reflects the plug-in estimator for CATE, now grounded in the multi-modal reasoning context.

### A.2 FURTHER ELABORATION ON KEY ISSUES IN THE MAIN PAPER

#### A.2.1 DETAILS OF RELAXING CERTAIN SELECTION CRITERIA

To ensure fairness and optimal performance, we strictly followed the data selection procedures outlined in the original s1 Muennighoff et al. (2025) and LIMO Ye et al. (2025) papers. This explains why the reported dataset sizes for these two methods are smaller in Table 1.

Specifically, the s1 aims to select a final set of 1,000 high-quality samples. It achieves this through a manual filtering step based on strict criteria. Hence, only 1,000 samples remain after this filtering. The LIMO adopts a multi-stage filtering process. It employs several advanced models to progressively

Table 5: Performance comparison of RAP and our implemented dynamic selection strategy RAP-Dynamic using Qwen2.5-VL-7b with the GRPO RL algorithm across three benchmarks.

| Method | Time (h) | MathVista | MMVet | LogicVista | Avg |
|---|---|---|---|---|---|
| Qwen2.5-VL-7b-RAP-Dynamic | 80.3 | 73.60 | 62.58 | 46.28 | 60.82 |
| **Qwen2.5-VL-7b-RAP (Ours)** | **52.80** | **73.20** | **63.31** | **46.53** | **61.01** |

select the most challenging samples. This stringent selection approach naturally results in fewer data samples compared to our method.

To avoid misunderstanding about performance drops from smaller datasets, we have conducted additional experiments by relaxing certain selection criteria for s1 and LIMO. For the s1, we replace the final manual selection of 1,000 samples with the top 6,103 samples ranked by difficulty score. For the LIMO, we relax the definition of a "difficult sample" from samples challenging for all models to those challenging for at least half of the models.

Consequently, in terms of the base model Qwen2.5-VL-7b, we expand the dataset sizes to 6,103 samples for s1 and 5,847 samples for LIMO (our RAP method uses 5,159 samples). As shown in the Table 1, increasing dataset sizes provides only marginal or negligible improvements compared to the original smaller datasets (s1: 1,000 samples; LIMO: 4,093 samples) reported in Table 1. The phenomenon with the motivation for data selection: indiscriminately enlarging datasets can introduce redundant or low-quality samples, diluting the value of cognitively rich data and limiting model improvement. Moreover, our RAP method continues to significantly outperform both s1 and LIMO, even after this data expansion. These results further confirm the effectiveness and fairness of our RAP method.

### A.2.2 DISCUSSION ABOUT DYNAMIC SELECTION STRATEGY

Exploring efficient dynamic selection strategies is a critical and promising direction for future research. As detailed in Section 4.3, we have investigated dynamic selection strategies to enhance model performance. Our preliminary experiments indicate that dynamically updating cognitive samples during training leads to only marginal improvements in performance. More importantly, this approach substantially increases computational complexity due to the repeated evaluation of the entire dataset across multiple training stages.

As shown in Table 5, our dynamic selection strategy yields slight improvements on just one dataset, while incurring a 52.1% increase in computational costs compared to the static RAP approach. Given our primary objective of balancing computational efficiency with model performance, we have opted for the static RAP selection paradigm. The RAP method achieves competitive or superior performance relative to training with the full dataset, while reducing computational overhead by over 43%. The rationale for not adopting the dynamic variant (RAP-Dynamic) lies in the observed gains being minor and inconsistent across datasets, coupled with the introduced significant computational overhead.

Future research will focus on developing lightweight adaptive data-sampling mechanisms that capture the evolving capabilities of the model without significantly increasing computational cost.

### A.2.3 ELABORATION ON THE PROPOSED DRM MODULE

To address the concern about the clarity of our DRM module, we provide further clarification below. First, we brief the roles of our proposed CDE and DRM. The primary objective of the CDE is to eliminate samples exhibiting a strong reliance on language priors, quantified by comparing the output discrepancy $D(x)$ between multi-modal and text-only predictions.

However, the CDE may unintentionally discard valuable multimodal samples, which exhibit low discrepancy scores due to inherent difficulty rather than reliance on language priors. These samples produce few correct multimodal predictions but consistently fail under text-only inputs, resulting in low discrepancy. Despite the low discrepancy, these are valuable for activating multimodal reasoning. To address this limitation, we propose the DRM to identify and reintroduce these informative discarded samples, thus preserving data complexity in training.

Table 6: Performance comparison of RAP and existing data selection methods using QwenVL2.5-7b with the GRPO RL algorithm across two new benchmarks.

| Method | POPE | VCR-wiki-EN |
|---|---|---|
| Qwen2.5-VL-7b-Full | 61.47 | 47.96 |
| Qwen2.5-VL-7b-s1 Muennighoff et al. (2025) | 86.03 | 65.59 |
| Qwen2.5-VL-7b-LIMO Ye et al. (2025) | 85.83 | 65.81 |
| **Qwen2.5-VL-7b-RAP (Ours)** | **86.21** | **66.45** |

Table 7: Performance comparison on two text-only benchmarks MATH500 Hendrycks et al. (2021) and AMC23 using Qwen2.5-7B and LLaMA3.1-8b, with GRPO as the RL algorithm. Random and Full refer to training with randomly sampled data and the complete dataset, respectively.

| Method | MATH500 | AMC23 |
|---|---|---|
| Qwen2.5-7b-Random | 72.40 | 50.25 |
| Qwen2.5-7b-Full | 74.40 | 51.25 |
| **Qwen2.5-7b-RAP (Ours)** | **74.60** | **51.75** |
| LLaMA3.1-8b-Random | 46.60 | 20.50 |
| LLaMA3.1-8b-Full | 48.60 | 22.75 |
| **LLaMA3.1-8b-RAP (Ours)** | **49.90** | **23.25** |

Specifically, during the CDE selection process, samples are discarded based on their output discrepancy score $D(x)$ as shown in Equation 4. However, certain discarded samples may be important for activating multimodal reasoning. For example, when setting the threshold is 0.3 (i.e., $\mu_c + \lambda_c \cdot \sigma_c = 0.3$) and the $M$ is set as 5, we observe scenarios where some samples produce few correct responses (1 correct out of 5) with multimodal inputs, while consistently failing all $M$ rollout outputs with text-only inputs (0 correct out of 5), i.e., $D(x) = 0.2 < \mu_c + \lambda_c \cdot \sigma_c$. As a result, these valuable yet challenging samples are excluded due to their low discrepancy scores. This inadvertently reduces the complexity of training data, potentially limiting model performance.

To address this issue, we propose the DRM to identify the challenging yet informative samples, satisfying $D_{\text{iff}} \in \left[\frac{1}{5}, 1\right)$ and $D(x) > 0$, which were inadvertently discarded by CDE due to their low discrepancy scores. The DRM subsequently reintroduces these discarded challenging samples by replacing overly simplistic samples previously retained by CDE (those with difficulty scores $D_{\text{iff}} = 0$).

Therefore, DRM effectively resolves the inherent limitations of CDE by preserving cognitively demanding instances. Through this complementary interaction, DRM maintains the diversity and complexity of the training data, significantly enhancing the model's multimodal reasoning capabilities.

To sum up, *the DRM does not contradict the purpose of CDE, instead, it addresses the limitation of CDE by retaining valuable multimodal difficult samples and preserving training set complexity*.

### A.3 KEY INSIGHT AND DISCUSSION

#### A.3.1 COMPARISON ON MORE DIVERSE MULTI-MODAL TASKS

To address concerns regarding the applicability of RAP to more complex scenarios, we conducted additional evaluations on more detailed vision-language benchmarks, including universal multimodal reasoning and VCR tasks. Specifically, we assess our model, QwenVL2.5-7b-RAP, on the challenging multi-modal datasets POPE Li et al. (2023) and the VCR task Zhang et al. (2025). As shown in Table 8, our proposed RAP method consistently outperforms existing methods across all evaluated metrics. These findings reinforce that RAP enhances performance in fine-grained vision-language tasks, demonstrating its robustness in complex, multi-modal scenarios.

#### A.3.2 GENERALIZATION ON TEXT-ONLY SCENARIOS

Our RAP method aims at enhancing multi-modal reasoning during the RL post-training phase. Therefore, the core CDE inherently depends on cross-modal signals, limiting its direct transfer to unimodal scenarios. To address potential concerns regarding applicability to LLMs, we adapt the ACE and DRM modules to text-only settings. In this adaptation, due to the absence of CDE, we

Table 8: Performance comparison between RAP and RAP-mid using QwenVL2.5-3b with the GRPO RL algorithm across three benchmarks. RAP-mid refers to the variant that utilizes intermediate (6th-layer) attention maps.

| Method | MathVista | MMVet | LogicVista | WeMath | Avg. |
|---|---|---|---|---|---|
| QwenVL2.5-3b-RAP-mid | 64.60 | 54.29 | **41.74** | 29.17 | 47.45 |
| **Qwen2.5VL-3b-RAP (Ours)** | **64.90** | **54.63** | 41.61 | **29.33** | **47.62** |

achieve the DRM by directly removing easy samples defined by $D_{\text{iff}} = 0$, *i.e.*, samples consistently answered correctly across all trials. This setup allows us to evaluate the effectiveness of our ACE and DRM in LLMs. Under this setting, we evaluated ACE and DRM on two widely used LLMs, Qwen2.5-7B and LLaMA3-7B, employing GRPO as the RL method. The results in the Table 7 show that ACE and DRM can achieve consistent performance gains in text-only contexts, confirming the broader generalizability of our RAP method. However, these improvements remain modest compared to the multi-modal scenario. We attribute this gap primarily to two reasons: (1) the original motivation and design of RAP cater to multi-modal contexts; (2) the absence of the crucial CDE component significantly reduces DRM's efficacy, as DRM aims to address the limitation of CDE.

### A.3.3 ANALYSIS ON THE CHOICE OF ATTENTION LAYERS

We clarify our rationale for using last-layer attention and provide additional experiments exploring attention maps from other layers.

As illustrated in Section 3.2 and Figure 2(b), attention-biased distributions typically manifest as excessively persistent high attention allocation across subsequent token sequences. Existing research Rohekar et al. (2023) has demonstrated that attention distributions in the final layer effectively capture causal reasoning behaviors. Such findings intuitively highlight the critical role of the last-layer attention maps in evaluating the reliability of the model's reasoning behavior. Inspired by this, we initially chose the last-layer attention distributions to effectively capture globally biased attention patterns that compromise genuine multi-modal reasoning.

However, we have conducted preliminary experiments to investigate the potential of Attention Confidence computed from intermediate layers. Recent work Yin et al. (2025) has indicated layers 5 to 8 within large language models play a critical role in multi-modal fusion. Therefore, we selected the 6th transformer layer's attention distribution as an alternative input for our ACE module and compared its efficacy against our original last-layer choice. Due to limited time, we utilized QwenVL2.5-3B as the base model.

The results presented in the following table demonstrate slight performance degradation when utilizing intermediate (6th-layer) attention maps. For most datasets, employing attention maps from the last layer consistently yielded superior performance. These results justify our choice of using last-layer attention maps.

### A.4 COMPARISON WITH STATE-OF-THE-ART METHODS

To further evaluate the robustness and effectiveness of the RAP approach, we augment our empirical analysis by incorporating three additional challenging evaluation datasets: *MMMU* Yue et al., and *ScienceQA* Lu et al. (2022). These datasets comprehensively probe the model's complex and universal multi-modal reasoning ability.

### A.4.1 MORE RESULTS USING THE BASE MODEL QWEN2.5-VL

**Results using the Qwen2.5-VL-7b.** As illustrated in Table 9, Qwen2.5-VL-7b trained with cognitive samples selected by our RAP method consistently surpasses the state-of-the-art approaches across all the benchmarks. Specifically, RAP achieves significant accuracy improvements of 1.1% on MMMU Yue et al. compared to the current state-of-the-art method LIMR Li et al. (2025)). These improvements demonstrate RAP's effectiveness in identifying high-quality cognitive samples, which can enhance the generalization and robustness of models in complex reasoning scenarios.

Table 9: Comparison with state-of-the-art methods. Experiments are conducted using the Qwen2.5-VL-3b Qwen et al. (2025) and Qwen2.5-VL-7b Qwen et al. (2025), employing GRPO as the RL method. "Time" denotes the total computation cost, including data selection and training. Bold font denotes the best result.

| Method | Sample | Time (h) ↓ | MMMU ↑ | ScienceQA ↑ |
|---|---|---|---|---|
| Qwen2.5-VL-7b | - | - | 45.33 | 83.02 |
| Qwen2.5-VL-7b-Full | 54,931 | 93.2 | 46.55 | 85.53 |
| Qwen2.5-VL-7b-s1 Muennighoff et al. (2025) | 1,000 | 55.9 | 45.78 | 84.57 |
| Qwen2.5-VL-7b-LIMO Ye et al. (2025) | 4,093 | 111.9 | 45.44 | 84.65 |
| Qwen2.5-VL-7b-LIMR Li et al. (2025) | 8,136 | 122.0 | 46.44 | 85.39 |
| **Qwen2.5-VL-7b-RAP (Ours)** | 5,159 | **52.80** | **47.56** | **85.87** |
| Qwen2.5-VL-3b | - | - | 43.56 | 80.71 |
| Qwen2.5-VL-3b-Full | 54,931 | 46.5 | 45.78 | 81.26 |
| Qwen2.5-VL-3b-s1 Muennighoff et al. (2025) (2025) | 1,000 | 38.4 | 44.22 | 80.63 |
| Qwen2.5-VL-3b-LIMO Ye et al. (2025) (2025) | 2,679 | 120.5 | 44.56 | 81.07 |
| Qwen2.5-VL-3b-LIMR Li et al. (2025) (2025) | 21,303 | 60.9 | 46.22 | 81.39 |
| **Qwen2.5-VL-3b-RAP (Ours)** | 4,374 | **32.0** | **46.78** | **81.51** |

Table 10: Comparison with state-of-the-art methods using LLaVA1.5-7b Liu et al. (2024) with the RL method GRPO, evaluating our RAP method on different base models.

| Method | MMVet | MathVista | LogicVista |
|---|---|---|---|
| LLaVA1.5-7b | 27.21 | 23.70 | 23.49 |
| LLaVA1.5-7b-s1 Muennighoff et al. (2025) | 28.03 | 24.00 | 23.62 |
| LLaVA1.5-7b-LIMO Ye et al. (2025) | 28.16 | 23.90 | 23.57 |
| LLaVA1.5-7b-LIMR Li et al. (2025) | 28.31 | 24.30 | 23.89 |
| **LLaVA1.5-7b-RAP (Ours)** | **28.45** | **24.50** | **24.16** |

### A.4.2 MORE RESULTS USING THE BASE MODEL QWEN2.5-VL

**Results using the Qwen2.5-VL-7b.** As illustrated in Table 9, Qwen2.5-VL-7b trained with cognitive samples selected by our RAP method consistently surpasses the state-of-the-art approaches across all the benchmarks. Specifically, RAP achieves significant accuracy improvements of 1.1% on MMMU Yue et al. compared to the current state-of-the-art method LIMR Li et al. (2025). These improvements demonstrate RAP's effectiveness in identifying high-quality cognitive samples, which can enhance the generalization and robustness of models in complex reasoning scenarios.

**Results using the Qwen2.5-VL-3b.** Table 9 presents improvements obtained by RAP using the Qwen2.5-VL-3b model, albeit less pronounced compared to those achieved with the 7b variant. Specifically, RAP demonstrates notable but comparatively modest accuracy gains. This performance discrepancy can be attributed to the reliance of our RAP method on the reasoning capabilities of the initial model for cognitive sample selection. Consequently, the inferior initial reasoning performance of the smaller-capacity Qwen2.5-VL-3b adversely impacts the effectiveness of selected training samples, thus limiting its performance enhancement potential relative to the larger-scale model.

### A.4.3 MORE RESULTS USING DIFFERENT BASE MODELS AND RL STRATEGIES

**Results using the LLaVA1.5-7b.** Table 10 presents the performance improvements achieved by our RAP method when applied to the LLaVA1.5-7b model Liu et al. (2024). Specifically, RAP consistently outperforms existing approaches across all evaluated metrics. These results not only underscore the efficacy of our approach in enhancing model performance but also demonstrate RAP's robust generalization across a variety of model architectures. Our RAP method's superior performance on three MMVet, MathVista, and LogicVista benchmarks further validates its potential to scale across different domains of multi-modal reasoning, reinforcing the versatility and effectiveness of RAP in practical applications.

**Results using the InternVL3-2b.** To further assess the effectiveness of RAP on various frameworks, we conducted experiments with InternVL3-2b trained using cognitive samples selected by RAP. As shown in Table 11, RAP consistently yields substantial performance gains, achieving gains of 1.3% and 0.6% on MMStar and LogicVista, respectively, compared to the model training with full data.

Table 11: Comparison with state-of-the-art methods using InternVL3-2b Chen et al. (2024b) with the RL method GRPO Guo et al. (2025), evaluating our RAP method across different model architectures.

| Method | MMStar | LogicVista |
|---|---|---|
| InternVL3-2b | 51.33 | 32.01 |
| InternVL3-2b-Full | 57.35 | 33.67 |
| InternVL3-2b-s1 Muennighoff et al. (2025) | 54.38 | 32.71 |
| InternVL3-2b-LIMO Ye et al. (2025) | 54.79 | 32.79 |
| InternVL3-2b-LIMR Li et al. (2025) | 57.64 | 33.85 |
| **InternVL3-2b-RAP (Ours)** | **58.64** | **34.29** |

Table 12: Comparison with state-of-the-art methods using Qwen2.5-VL-7b Qwen et al. (2025) with the RL method RLOO Ahmadian et al. (2024), evaluating our RAP method on different training datasets and RL algorithms.

| Method | MMStar | LogicVista |
|---|---|---|
| Qwen2.5-VL-7b | 56.07 | 44.52 |
| Qwen2.5-VL-7b-Full | 61.47 | 47.96 |
| Qwen2.5-VL-7b-s1 Muennighoff et al. (2025) | 59.62 | 46.23 |
| Qwen2.5-VL-7b-LIMO Ye et al. (2025) | 59.89 | 46.55 |
| Qwen2.5-VL-7b-LIMR Li et al. (2025) | 61.13 | 47.61 |
| **Qwen2.5-VL-7b-RAP (Ours)** | **62.27** | **48.10** |

These empirical results underscore the universality and architecture-agnostic capability of RAP in selecting informative cognitive samples, thus effectively boosting model performance across different frameworks.

**Results using the RLOO.** To demonstrate RAP's generalization capability across different RL strategies and training datasets, we further integrate RAP with the Reinforce Leave-One-Out (RLOO) method Ahmadian et al. (2024) using the Mulberry-10k dataset. As illustrated in Table 12, RAP combined with RLOO consistently exceeds alternative data selection strategies. These findings confirm RAP's flexibility in accommodating distinct RL algorithms and diverse training datasets, further emphasizing its broad applicability and robustness across different experimental conditions.

## A.5 IMPLEMENTATION DETAILS

### A.5.1 THE OVERVIEW OF RAP

Algorithm 1 provides a detailed overview of our RAP framework, depicting its three-step data selection pipeline. Initially, the Causal Discrepancy Estimator (CDE) identifies and eliminates language-prior biased samples by evaluating output discrepancies between text-only and multi-modal inputs. Subsequently, the Attention Confidence Estimator (ACE) further filters attention-biased samples based on token-level self-attention distributions. Lastly, the Difficulty-aware Replacement Module (DRM) strategically replaces trivial samples with cognitively challenging alternatives, ensuring robust data complexity to elevate multi-modal reasoning capabilities.

### A.5.2 DETAILS OF TOTAL TIME

Figure 8 and 9 provide a comprehensive analysis of computational resources required by various data selection methodologies and subsequent model training across different sample sizes. We systematically evaluate five comparative approaches using the Qwen2.5-VL-7B and Qwen2.5-VL-3B models as specified in Table 9: Full data (using the complete MM-Eureka dataset Meng et al. (2025)), s1 Muennighoff et al. (2025), LIMO Ye et al. (2025), LIMR Li et al. (2025), and our proposed RAP method. Each step's rationale, process, and respective computational cost are detailed below, with black indicating the 7B variant and blue for the 3B variant.

**(1) Full Data:**

1. **Complete Dataset Training (93.2 hours; 46.5 hours)**: In this baseline setting, RL training is directly performed on the entire MM-Eureka dataset, encompassing a full epoch consisting of 107 training steps. It ensures comprehensive exposure to all available data.

---

**Algorithm 1** Reasoning Activation Potential (RAP)-based Data Selection and Training

---

Original dataset $D = \{(x_t^i, x_v^i, y^i)\}_{i=1}^N$, pretrained model $\theta$, output discrepancy threshold $\lambda_c$, attention confidence threshold $\lambda_a$, training steps $T$ Curated cognitive samples $x_{cd}$ **Stage 1: Causal Discrepancy Estimator (CDE)** each sample $(x_t, x_v, y) \in D$ Generate $M$ outputs separately using multimodal $(x_t, x_v)$ and text-only $(x_t)$ inputs Compute output consistency metric:

$$D(x) = \frac{1}{M} \sum_{j=1}^{M} \left[ \mathbb{I}(Y_1^{(j)} = y) - \mathbb{I}(Y_0^{(j)} = y) \right]$$

Form discrepancy-filtered dataset:

$$X_{\text{cde}} = \{ x \mid D(x) \geq \mu_D + \lambda_c \sigma_D \}$$

**Stage 2: Attention Confidence Estimator (ACE)** Initialize confidence-filtered set $X_{\text{ace}} = \emptyset$ each sample $x \in X_{\text{cde}}$ Perform forward pass and extract self-attention matrix $A \in \mathbb{R}^{L \times L}$ from the final layer Compute attention confidence:

$$\psi_j(A) = \prod_{i=j}^{L} (\sigma \cdot A_{i,j}), \quad \psi(A) = \max_j \psi_j(A)$$

$\psi(A) \leq \lambda_a$ Add sample $x$ to $X_{\text{ace}}$ **Stage 3: Difficulty-aware Replacement Module (DRM)** Compute difficulty score for each sample $x \in X_{\text{ace}}$:

$$D_{\text{diff}}(x) = 1 - \frac{1}{M} \sum_{j=1}^{M} \mathbb{I}(\text{output}^{(j)} \text{ is correct})$$

Identify trivial samples:

$$X_{\text{easy}} = \{ x \mid D_{\text{diff}}(x) = 0 \}$$

Select top $|X_{\text{easy}}|$ challenging samples from $D \setminus X_{\text{ace}}$ based on descending difficulty, forming the challenging set $X_{\text{hard}}$ Construct final cognitive dataset:

$$x_{cd} = (X_{\text{ace}} \setminus X_{\text{easy}}) \cup X_{\text{hard}}$$

**Stage 4: Cognitive Dataset Training** Training model $\theta$ on cognitive samples $x_{cd}$ via reinforcement learning algorithms (GRPO) for $T$ steps

---

**(2) s1 Muennighoff et al. (2025):**

1. **API-based Preprocessing (1.0 hour; 1.0 hours)**: Cleans the initial data by removing low-quality entries (*e.g.*, incomplete, ambiguous samples) and standardizing metadata, ensuring dataset consistency prior to deeper analysis.

2. **Two-round Dual-Model Difficulty Assessment (9.5+9.5 hours; 9.5+9.5 hours)**: In this phase, we conduct a comprehensive difficulty evaluation utilizing two distinct and independently trained models. Each model individually assesses the difficulty level of every candidate question, consuming approximately 9.5 hours (5.2 hours) per model. Questions that both models successfully solve are systematically excluded, as their uniformly correct predictions indicate trivial complexity or minimal informative value. Consequently, this dual-model filtering strategy effectively retains only challenging or ambiguous samples, significantly enhancing dataset informativeness and reducing redundancy.

3. **Final Annotation and Sampling (1.0 hour; 1.0 hours)**: The refined subset subsequently undergoes annotation through an API-based labeling process. This procedure assigns explicit complexity or domain labels to each remaining sample, ensuring balanced representation and coverage of diverse reasoning scenarios. The resulting dataset thus maximizes representational diversity and maintains a high degree of relevance and informational richness.

4. **Training Phase (34.9 hours; 17.4 hours)**: Training with the curated data subset for 40 steps.

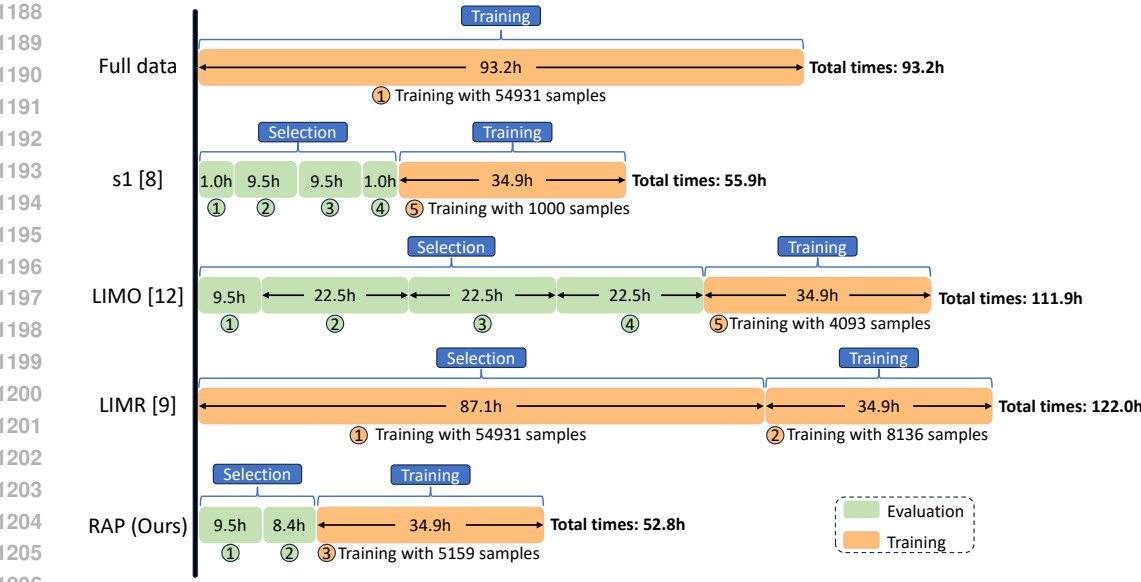

Figure 8: The total computational time required by different data selection and training with various amounts of samples using the Qwen2.5-VL-7b.

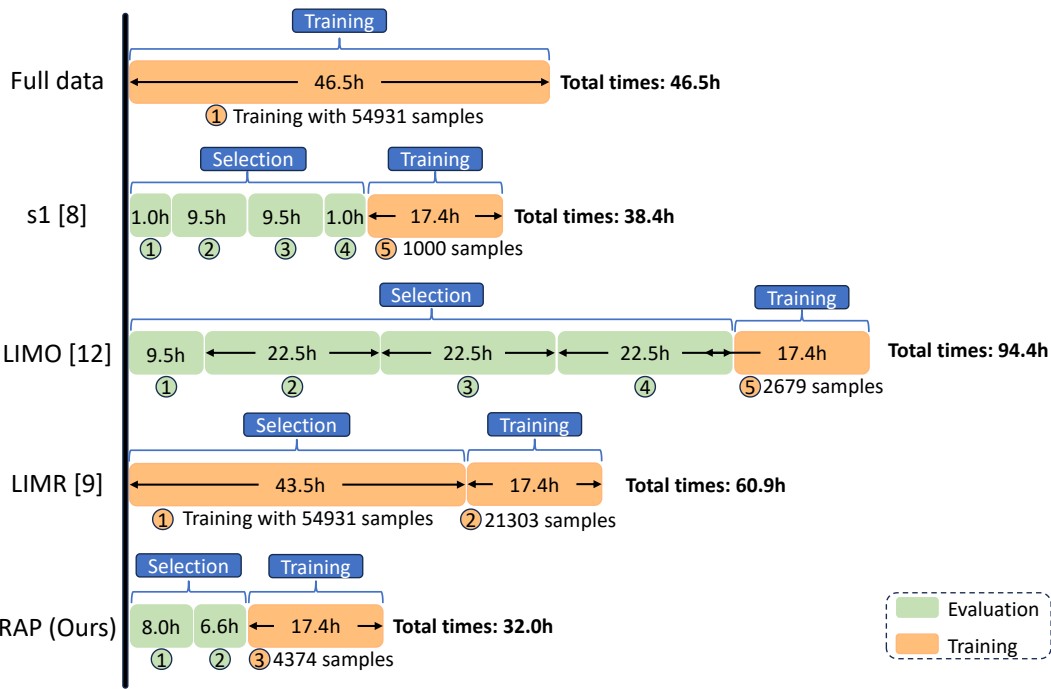

Figure 9: The total computational time required by different data selection and training with various amounts of samples using the Qwen2.5-VL-3b.

**(3) LIMO Ye et al. (2025):**

1. **Initial Filtering via Qwen2.5-Instruct (9.5 hours; 9.5 hours):** This stage utilizes Qwen2.5-Instruct to efficiently remove obviously trivial or low-complexity samples. By rapidly discarding these samples at an early stage, computational resources are strategically conserved, allowing subsequent analyses to focus solely on potentially challenging or informative examples.

2. **Multi-Model Validation – Stage 1 (22.5 hours; 22.5 hours)**: In the first validation stage, we employ InternVL3-38B to independently evaluate the filtered dataset over five iterative passes (4.5 hours per pass), emphasizing detection of semantically ambiguous samples and annotation noise. This phase establishes a foundational layer of quality assurance by leveraging InternVL's fine-grained multimodal reasoning capabilities.

3. **Multi-Model Validation – Stage 2 (22.5 hours; 22.5 hours)**: The second stage invokes Qwen2.5-VL-32B, which further refines the dataset by validating samples previously accepted or contested in Stage 1. Through the same 5-pass procedure (totaling 22.5 hours), Qwen2.5 offers complementary linguistic priors and instruction-following precision to enhance sample reliability.

4. **Multi-Model Validation – Stage 3 (22.5 hours; 22.5 hours)**: Finally, LLaVA-OneVision conducts an independent pass-through using visual-textual alignment cues. This stage not only reinforces agreement across modalities but also uncovers visually grounded inconsistencies that may elude text-centric evaluation. The redundant yet diverse model ensemble ensures a high-confidence consensus on challenging data points.

5. **Training Phase (34.9 hours; 17.4 hours)**: Training with the curated data subset for 40 steps.

**(4) LIMR Li et al. (2025):**

1. **Learning Impact Measurement-based Selection (87.1 hours; 43.5 hours):** This phase involves conducting reinforcement learning (RL) training over one complete epoch (107 training steps) on the initial dataset of 54,931 samples. Throughout this epoch, LIMR systematically records and evaluates each sample's contribution to the learning trajectory, assigning higher priority to samples demonstrating greater alignment and impact on model performance. By quantitatively measuring these dynamics, the approach efficiently identifies a reduced subset of 8,136 high-value examples, optimizing subsequent resource allocation and computational efficiency.

2. **Training Phase (34.9 hours; 17.4 hours):** Utilizing the rigorously curated subset of 8,136 samples, LIMR conducts focused RL training over 40 training steps. This strategically condensed training phase capitalizes on the carefully selected dataset to maximize the model's reasoning potential.

**(5) RAP (Ours):**

1. **Reasoning Activation Potential-based Selection (17.9 hours; 14.6 hours):** Our proposed method initially utilizes text-only inputs to assess the entire dataset (**8.4 hours**; 8.0 hours), followed by multimodal inputs to compute output-level distribution and self-attention values (**9.5 hours**; 6.6 hours). Subsequently, sample selection is based on discrepancies in outputs and attention metrics, complemented by the DRM module to replace overly simple samples with cognitively more challenging ones.

2. **Training Phase (34.9 hours; 17.4 hours)**: Training with the curated data subset for 40 steps.

A.6    QUALITATIVE ANALYSIS

In this section, we conduct a qualitative evaluation by visualizing cognitive samples and comparing the reasoning processes of our proposed RAP approach with the state-of-the-art LIMR method Li et al. (2025).

A.6.1    VISUALIZATION OF COGNITIVE SAMPLES

As demonstrated in Figure 10(a) and 11(a), we illustrate cognitive samples selected by our RAP approach, highlighting their effectiveness in integrating geometric and logical reasoning across modalities. Specifically, in the first example (top figure), RAP successfully leverages multi-modal inputs to accurately compute the area of triangle $\triangle BDC$ by correctly identifying that side $AD$ acts as the triangle's height, a critical detail overlooked in a text-only approach. The second example (bottom figure) further underscores RAP's capability to identify corresponding angles and properly leverage

geometric rules (such as the parallel line theorem) using visual modalities, leading to the accurate calculation of angle measurements. These cases collectively illustrate two essential characteristics of

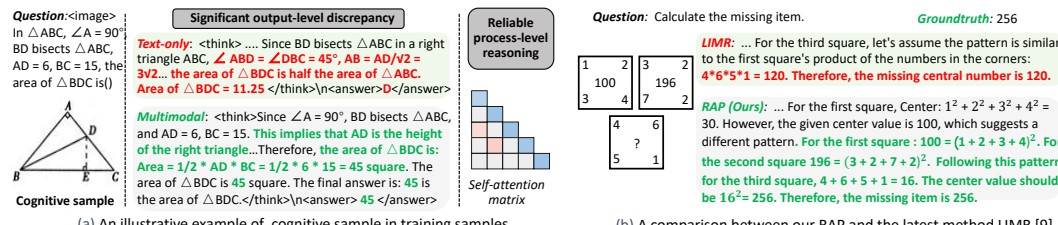

(a) An illustrative example of cognitive sample in training samples       (b) A comparison between our RAP and the latest method LIMR [9]

Figure 10: Illustration of (a) characteristics of cognitive samples selected by our RAP method and (b) Comparison of the reasoning processes between our RAP method and the state-of-the-art LIMR Li et al. (2025).

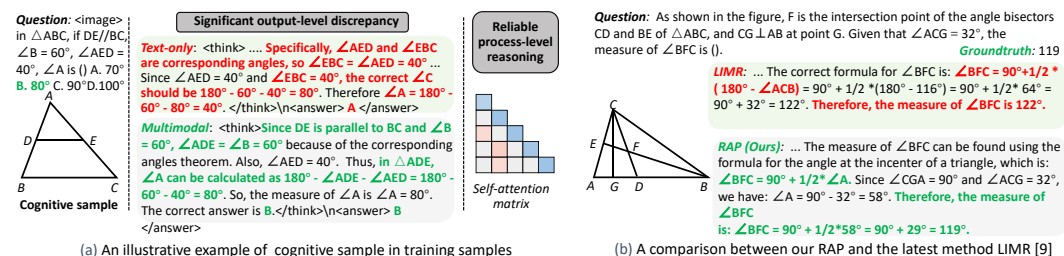

(a) An illustrative example of cognitive sample in training samples       (b) A comparison between our RAP and the latest method LIMR [9]

Figure 11: Illustration of (a) characteristics of cognitive samples selected by our RAP method and (b) Comparison of the reasoning processes between our RAP method and the state-of-the-art LIMR Li et al. (2025).

cognitive samples selected by RAP: (a) they demonstrate the necessity and effectiveness of integrating multi-modal information by revealing significant reasoning discrepancies between multi-modal and purely textual methods; and (b) they prioritize meaningful geometric details, avoiding distractions by irrelevant textual tokens, thereby optimizing the reasoning process.

### A.6.2 COMPARISON CASE ANALYSIS

Further, as illustrated in Figure 10(b) and 11(b), we provide a detailed comparison between the reasoning processes of RAP and the state-of-the-art LIMR approach. In the first scenario (top figure), LIMR incorrectly assumes a simplistic numerical multiplication pattern, resulting in a miscalculation of the missing central number. In contrast, RAP identifies the accurate multi-modal reasoning pattern—namely, the summation of squared values of surrounding numbers—achieving the correct central value. Similarly, in the second scenario (bottom figure), LIMR misapplies angle bisector formulas and fails to correctly leverage multi-modal geometric constraints, thus deriving an incorrect angle measurement. Conversely, RAP appropriately incorporates geometric principles (such as angle bisector rules and interior angles relationships) and multi-modal integration to determine the accurate angle measure.

These comparison cases robustly demonstrate that cognitive samples selected through RAP significantly enhance model performance by encouraging precise and reliable reasoning using multi-modal geometric information. Consequently, RAP-trained models exhibit superior interpretability and accuracy over methods lacking effective integration across modalities.

### A.7 REPRODUCIBILITY STATEMENT

We are committed to ensuring that the research presented in this paper is reproducible. All relevant details, including the experimental setup, model architectures, hyperparameters, and datasets, are fully documented in the main text and supplementary materials. For reproducibility, we provide the following resources: (1) a link to the anonymized source code for our method, which includes

all necessary scripts for reproducing the results reported in the experiments; (2) a comprehensive description of the datasets used, including any data processing steps undertaken to prepare the data for experimentation; (3) detailed explanations of the theoretical assumptions underlying our method, as well as proofs of key claims provided in the appendix. We also describe the evaluation metrics used to assess our method's performance, ensuring that others can replicate our experiments and verify the results. By making these materials publicly available, we hope to facilitate transparency and promote further validation of our approach within the community.

## A.8   ETHICS STATEMENT

This work adheres to the ICLR Code of Ethics, and we have taken all necessary precautions to ensure that our research meets ethical standards. The experiments in this paper do not involve human subjects, sensitive data, or any forms of direct interaction with individuals. Our data is derived from publicly available sources, and all data processing steps have been conducted in compliance with applicable privacy regulations. We explicitly acknowledge that our method, while improving multi-modal reasoning, does not introduce any discriminatory biases or unfairness across different demographic groups. The authors have no financial or non-financial interests that could have influenced the results or interpretation of this work.

## A.9   FUTURE WORK

While our proposed RAP method demonstrates significant improvements in both training efficiency and model performance by pre-selecting high-value samples prior to RL training, it is fundamentally a static data selection strategy. Specifically, the cognitive driver subset is identified using the initial model state before RL fine-tuning, and remains fixed throughout training. As a result, the method may not fully adapt to the evolving learning dynamics or sample difficulty distribution during RL training. Our preliminary exploration (see Figure 7(d)) shows that dynamic data selection, i.e., re-evaluating and updating the cognitive driver set at intermediate stages, can further improve model performance. However, such dynamic strategies introduce additional computational overhead due to repeated screening of the entire dataset, which may offset the efficiency gains brought by sample reduction. In future work, we aim to develop more efficient dynamic data selection mechanisms, such as periodic or adaptive sampling, that balance computational cost and model performance, enabling truly adaptive curriculum learning during RL post-training.

## A.10   DECLARATION OF LLM USAGE

The LLM is utilized for writing and refinement purposes. The paper proposes a novel data selection paradigm for efficient multi-modal reasoning in multi-modal large language models (MLLMs), which serve as the experimental backbone.

