# OpenReview forum: "Truth in the Few: High-Value Data Selection for Efficient Multi-Modal Reasoning"
_ICLR.cc/2026/Conference — ICLR 2026 Conference Withdrawn Submission_

### Official Review · Reviewer_Lh2x · 2025-10-30

**Soundness:** 3
**Presentation:** 3
**Contribution:** 1
**Rating:** 2
**Confidence:** 4

**Summary:**

This paper, addresses the significant computational costs and data redundancy associated with using extensive training data in the reinforcement learning post-training phase to enhance the complex reasoning abilities of multi-modal large language models.

- Core Thesis: The authors challenge the prevailing assumption that large-scale data is necessary for improving multi-modal reasoning. They reveal a "truth in the few" phenomenon: smaller, high-quality datasets, termed cognitive samples, can match or surpass the performance achieved by training on the full corpus.

- Key Methodology: The paper proposes a novel data selection paradigm called Reasoning Activation Potential (RAP). RAP aims to identify these cognitive samples that effectively stimulate genuine multi-modal reasoning. The method comprises three complementary components:
  - Causal Discrepancy Estimator (CDE): Based on the Potential Outcome Model (POM), CDE estimates the causal effect of the visual modality on model predictions. It eliminates samples where the output discrepancy is low (meaning predictions are nearly the same with or without the image), effectively filtering out language-prior biased samples.
  - Attention Confidence Estimator (ACE): This estimator assesses the quality of the internal reasoning process by exploiting token-level self-attention distributions from the final transformer layer. It is designed to discard attention-biased samples characterized by over-emphasis on semantically irrelevant tokens.
  - Difficulty-aware Replacement Module (DRM): To counteract the accidental discarding of challenging but valuable samples by CDE/ACE, DRM replaces overly simplistic samples with cognitively challenging alternatives. This ensures sufficient data complexity to enhance the model's reasoning upper bound.

- Main Contributions and Findings:
  - Efficiency and Performance : Experiments across six datasets show that RAP consistently achieves superior performance while using only a sparse subset (e.g., 9.3% of training data) and reducing computational costs by over 43% (e.g., Qwen2.5-VL-7b model, from 54,931 samples/93.2h to 5,159 samples/52.8h).
  - Novel Methodology: The introduction of the CDE and ACE for principled bi-modal quality assessment, combined with the DRM for maintaining data complexity, represents a significant methodological contribution.
  - Generalizability : The RAP method demonstrates robust applicability and consistent superiority across different base models (Qwen2.5-VL-3B/7B, InternVL3-2b, LLaVA1.5-7b) and various RL algorithms (GRPO, RLOO)

**Strengths:**

1. The paper provides strong validation and in-depth analysis of the "Less is More" principle within the domain of Multi-modal Large Language Models. Through empirical analysis, the paper finds that training with only 20% of the data leads to merely a 0.8% performance degradation compared to the full dataset, leading to the proposal of the "truth in the few" phenomenon. Experimental results demonstrate that the method achieves superior performance compared to models trained on the full corpus, while reducing computational costs by over 43%, effectively balancing performance and efficiency.
2. The work proposes a novel and effective multi-modal data selection paradigm called Reasoning Activation Potential (RAP). The RAP paradigm aims to identify "cognitive samples" that effectively trigger the model's multi-modal reasoning capability. It innovatively evaluates the reasoning activation potential of samples from two complementary perspectives: output-level reasoning discrepancy and process-level reasoning confidence. The Causal Discrepancy Estimator (CDE), based on the Potential Outcome Model (POM), eliminates samples overly reliant on language priors by quantifying the causal effect of the visual modality. The Attention Confidence Estimator (ACE) filters out attention-biased samples by assessing the quality of the internal reasoning process via token-level self-attention distribution. Additionally, the Difficulty-aware Replacement Module (DRM) is introduced to address the limitation of CDE and ACE possibly discarding valuable yet challenging samples, thereby ensuring sufficient training data complexity and raising the model's reasoning upper bound. The ablation study confirms the positive contribution of each component (CDE, ACE, and DRM) to the final performance gain. The RAP method shows robust performance gains and generalizability across various base models (Qwen2.5-VL-3b/7b, InternVL3-2b, LLaVA1.5-7b) and different Reinforcement Learning algorithms (GRPO, RLOO).

**Weaknesses:**

1. Risk of Insufficient Novelty
  - The paper's core finding, "less is more," is not an entirely new concept. Previous works in the context of large language models (LLMs), such as s1and LIMO, have already proposed the idea of using data selection to enhance reasoning performance and reduce training costs.
  - Although the concept is extended to the multi-modal domain in this work, the proposed Causal Discrepancy Estimator (CDE) is fundamentally a measure of output prediction difference between multi-modal and text-only inputs. This approach shares a similar core idea with existing methodologies that address "language-prior bias" or "object hallucination" by contrasting multi-modal and text-only model responses.
2. Overly Complex and Coupled Methodology Design
  - The RAP paradigm relies on the sequential collaboration of three components: CDE, ACE, and DRM. This complex pipeline introduces a higher number of hyperparameters, specifically $\lambda_{c}$ and $\lambda_{a}$, thereby increasing the difficulty of tuning the method and its implementation cost.
  - The Attention Confidence Estimator (ACE) is particularly problematic as it relies on analyzing the self-attention matrix from the final transformer layer. This tight coupling of the data selection process to a specific internal mechanism of the model architecture (the transformer's attention layer) compromises the theoretical generality and ease of application across different model frameworks.
3. Lack of Sufficient Theoretical Justification and Intrinsic Rationality for Data Filtering
  - Limitations of CDE: CDE identifies language-prior biased samples by discarding those with a small output discrepancy $D(x)$. However, samples that are highly challenging might have few correct multi-modal predictions (e.g., only 1 correct out of $M$ rollouts) but consistently zero correct text-only predictions. In this case, $D(x)$ may still be low (e.g., $D(x)=0.2$ for $M=5$). Directly eliminating these challenging but valuable samples, which could contribute to improving the model's reasoning upper bound, lacks a robust theoretical justification, especially since the DRM component is later introduced to address this specific limitation.
  - Limitations of ACE: ACE aims to filter out "attention-biased samples" based on attention distributions. The phenomenon of "attention bias" (over-attending to irrelevant tokens) is arguably a defect inherent to the current state of the model $\theta$ (its internal reasoning dynamics), rather than an intrinsic property of the data sample itself. Training a model by solely removing samples that expose its current flaws might lead to a training data distribution biased away from challenging real-world scenarios, thereby evading the root cause of the model's internal reasoning shortcomings.
4. Relatively Limited Scope of Experimental Evaluation
  - Although supplementary results for tasks like POPE and VCR are provided in the Appendix, incorporating a more diverse and heterogeneous set of modern datasets into the primary training and evaluation pipeline would more strongly support the versatility and generalizability of the RAP framework.
5. Inappropriate Selection and Comparison with Baselines
  - The paper uses s1 and LIMO as primary baselines. However, both s1 and LIMO were originally developed and evaluated for data selection during the Supervised Fine-Tuning (SFT) phase.
  - In contrast, RAP and its evaluation focus on data selection for the Reinforcement Learning (RL) post-training phase.
  - The RL phase has fundamentally different optimization objectives and loss functions compared to SFT. Directly comparing a method designed for RL (RAP) with methods designed for SFT (LIMO and s1) is methodologically unsound. The observed poorer performance of LIMO and s1 in the RAP evaluation is likely due to this task mismatch, rather than a genuine invalidity of their core SFT data selection principles.

**Questions:**

- Can the efficiency of the proposed method be proven theoretically?
- Consider adding more diverse benchmarks for future evaluation, for instance:
  - BMMR: A Large-Scale Bilingual Multimodal Multi-Discipline Reasoning Dataset
  - CMMU: A benchmark for Chinese multi-modal multi-type question understanding and reasoning.
  - ScienceQA: A novel resource for question answering in scholarly articles.
  - R-Bench: Graduate-level multi-disciplinary benchmarks for LLM & MLLM complex reasoning evaluation.

---

### Official Review · Reviewer_zsCP · 2025-10-30

**Soundness:** 3
**Presentation:** 3
**Contribution:** 3
**Rating:** 6
**Confidence:** 3

**Summary:**

The authors propose RAP (Reasoning Activation Potential)—a pre-training data selection pipeline that keeps only cognitive samples, i.e., examples that truly trigger multi-modal reasoning.
RAP has three parts: CDE (Causal Discrepancy Estimator) filters out items solvable by text-only priors by contrasting MLLM outputs with and without images. ACE (Attention Confidence Estimator) removes items whose intermediate reasoning is hijacked by irrelevant tokens (spurious attention). DRM (Difficulty-aware Replacement Module) re-injects challenging examples so the kept set isn’t too “easy.”

**Strengths:**

* Moves beyond generic “hard example mining” to explicitly target multi-modal activation (CDE) and weed out spurious process signals (ACE).
* Practical efficiency: about 7–10% of data surpasses full-data RL, very attractive for teams constrained by GPU budgets.
* Improvements hold across multiple datasets, two model sizes, and more than one RL algorithm.
* CDE/ACE contribute complementary gains; DRM prevents the curated set from collapsing into “too easy,” addressing an important ceiling effect.

**Weaknesses:**

* ACE uses a length-multiplicative attention score without proper normalization. The attention confidence is effectively a product over a token run, e.g., $ \psi_j(A)=\prod_{i=j}^{L} (\sigma \cdot A_{i,j}) $. This formulation couples score magnitude to reasoning-chain length and can explode/vanish with longer sequences; it also mixes scale from $\sigma$ into the multiplicative path, making thresholding brittle. A log-domain or normalized aggregation (e.g., mean/log-sum-exp over a fixed window) would likely be more length-robust.

* Fixed global thresholds with limited cross-task validation. The paper fixes $(\sigma,\lambda_c,\lambda_a)$ to $(2.0, 0.5, 0.1)$ and mainly tunes $\lambda_c,\lambda_a$ on a single dataset, leaving unclear whether these settings transfer across models/tasks or just capture one distribution’s quirks.

* As shown in Table 2,3,4 Several reported wins are within a few points. Without robust multi-seed reporting on each benchmark and configuration, it’s hard to judge if gains exceed running noise. A 3–5 seed protocol with fixed compute would clarify statistical significance.

* The overall pipeline’s benefit depends on $(\sigma,\lambda_c,\lambda_a,M,k)$ choices and their interactions. The paper doesn’t yet demonstrate that these are pool-stable across datasets, model-stable across backbones variants, and compute-fair (same budget under different settings).

**Questions:**

Please refer to the Weaknesses.

---

### Official Review · Reviewer_bWfR · 2025-11-03

**Soundness:** 3
**Presentation:** 3
**Contribution:** 2
**Rating:** 4
**Confidence:** 3

**Summary:**

The paper introduces the Reasoning Activation Potential (RAP) paradigm, challenging the notion that extensive data is necessary for post-training Multi-modal Large Language Models (MLLMs). RAP efficiently selects a small set of "cognitive samples" by using two estimators: the Causal Discrepancy Estimator (CDE), which filters out language-prior biased samples, and the Attention Confidence Estimator (ACE), which discards samples with irregular attention in intermediate reasoning steps. Complemented by a Difficulty-aware Replacement Module (DRM) to maintain dataset complexity, RAP achieves superior multi-modal reasoning performance across six datasets while utilizing only 9.3% of the training data and cutting computational costs by over 43%.

**Strengths:**

1. Interesting Problem: The approach addresses a compelling and relevant challenge.
2. Intuitive Approach: The methodology presents a clear, logical progression that is easy to follow and understand.
3. Well-Presented Work: The presentation of the work is thorough and well-organized

**Weaknesses:**

1. "However, existing data selection methods rely on unimodal textual quality, such as human-annotated difficulty estimation" - I think human annotation also has multimodal data. Why is there an emphasis on data selection methods relying on unimodal data quality?

2. The discussion of data selection for reasoning in the related works section lacks sufficient details on prior methods, and it repeatedly highlights the need for data selection without thoroughly addressing existing approaches. It would be beneficial to reference methods such as meta-learning techniques for data selection, which could offer a more comprehensive perspective on this issue.

3. For the ACE, why is self-attention used and not some form of cross-attention, which would be more in line with the spirit of the CDE?

4. Clarify the criteria used for attention-based filtering — specifically, how irregular attention is quantified, what threshold ((\lambda_a)) is applied, and how this value is determined or tuned.

5. "First, we apply RAP to select cognitive samples using the initial model without any training." Identification of cognitive samples are done using a model that has not undergone GRPO. I think the selected samples might have a bias (or sub-optimal)? Bi-level optimization (BLO) based methods address this better, where in first stage model is fine-tuned and second stage samples are selected. This is performed iteratively. Ideally, BLO is designed to solve such a nested problem where one stage cannot be solved without solving other (selecting samples is needed to fine-tune model, however fine-tuned model give better guidance on what samples to select).

6. Please provide more details on the methods being compared against (baseline methods) in section 4.1. This understanding is needed for the inference of the results section. Also, why no meta-learning based method is compared to, for instance, https://arxiv.org/pdf/2505.20241(similar methods that do this).

7. L308 – what does 9.5% or 7.9% mean? Which two entities are being referred to here?

8. "Moreover, the comparison between No.2 and No.3 demonstrates that the DRM can further refine the reasoning performance by replacing easy samples with more appropriate hard samples." - compares ACE VS (CDE AND DRM). How does it lead to that conclusion?

9. In the ablation study, to better understand CDE, I suggest using samples that rely solely on language priors, rather than those combining visual and textual inputs, and conducting experiments that compare combinations of ACE, DRM, and the complement of CDE for comparison.

10. What is the y axis in figure 5 c? What is cross-modal reasoning utilization?

11. At L414, validate the qualitative claim in Figure 6(b)—that the model leverages multimodal features from cognitive samples—by quantitatively evaluating it. This can be done by removing the CDE component, selecting only language-only samples, and measuring the corresponding metric (see point 9).

**Questions:**

Please see above.

---

### Official Review · Reviewer_A1Bu · 2025-11-08

**Soundness:** 3
**Presentation:** 2
**Contribution:** 2
**Rating:** 4
**Confidence:** 3

**Summary:**

This paper proposes Reasoning Activation Potential (RAP), a data selection framework for efficient post-training of multi-modal large language models (MLLMs). It identifies cognitive samples—the small subset of data that truly triggers multi-modal reasoning—using two estimators: the Causal Discrepancy Estimator (CDE) to filter samples relying on language priors, and the Attention Confidence Estimator (ACE) to discard those dominated by irrelevant tokens. A Difficulty-aware Replacement Module (DRM) further ensures training complexity. Experiments on six datasets show RAP achieves superior reasoning performance with only 9.3% of the data and reduces computation by 43%.

**Strengths:**

The paper presents an innovative approach to data efficiency by identifying cognitive samples, effectively challenging the belief that large-scale datasets are necessary for strong multi-modal reasoning.
The proposed CDE and ACE modules are theoretically grounded and interpretable, offering clear insights into how data contributes to reasoning improvement.
Experiments show strong empirical results, achieving comparable or better performance with only 9.3% of data and significantly reducing computational cost by over 43%.

**Weaknesses:**

- The innovation of the Output-level Discrepancy Calculation is limited. Using causal inference to evaluate whether multi-modal samples contain language priors has been explored in prior work, such as [1] Counterfactual VQA: A Cause-Effect Look at Language Bias and [2] Counterfactual Reasoning for Out-of-distribution Multimodal Sentiment Analysis.


- The paper lacks clear definitions for causal inference concepts, which makes the logical flow less coherent. For example, what is the formal definition of the Individual Treatment Effect (ITE)? How does it differ from the direct effect and indirect effect—or is it a specific type of them?


- It is unclear why Y₀(u) can be represented simply by removing the image input. Should Y₀(u) correspond to an input without the image, or rather to an image containing completely unbiased content?


- Equation (4) requires further explanation—why can it represent discrepancy, and what is the high-level definition of this discrepancy? What exactly is being measured or contrasted?


- The Difficulty-aware Replacement Module (DRM) may unintentionally introduce samples with language priors, which could counteract the goal of bias reduction.


Minor comment: The algorithmic flowchart is crucial for understanding the proposed method and should be included in the main text. Some symbols and notations (e.g., dataset representations) should also be explicitly defined in the paper body rather than only in the appendix.

**Questions:**

See weakness

---

### Official Review · Reviewer_B8V2 · 2025-11-09

**Soundness:** 3
**Presentation:** 3
**Contribution:** 2
**Rating:** 4
**Confidence:** 3

**Summary:**

The paper proposes RAP (Reasoning Activation Potential), a data-selection framework for multimodal RL post-training of MLLMs. RAP aims to identify “cognitive samples” that truly stimulate multimodal reasoning, rather than language-prior–biased or attention-biased examples. It does so via two estimators—CDE, which measures output discrepancy between multimodal and text-only predictions under a potential-outcome view, and ACE, which flags samples with low-confidence or misfocused self-attention—and a DRM module that reintroduces challenging but filtered samples. On MM-Eureka and a Mulberry subset with Qwen2.5-VL and InternVL3 backbones, RAP outperforms full-data RL and prior selection methods (s1, LIMO, LIMR) using only ~8–10% of the data and substantially less wall-clock time.

**Strengths:**

The paper addresses a practically important question for multimodal RL: how to identify and exploit the relatively small subset of examples that genuinely drive multimodal reasoning, rather than spending compute on language-prior–biased or trivial samples. The RAP framework is conceptually appealing in that it decomposes “reasoning activation” into three complementary dimensions: causal discrepancy (CDE, grounded in the potential outcome model), attention confidence (ACE, using final-layer self-attention), and difficulty-aware replacement (DRM, to reintroduce hard but informative examples). This gives a structured and interpretable view of which samples matter most for training. Empirically, the evaluation is thorough within the chosen scope. RAP is tested on multiple multimodal reasoning benchmarks (MathVista, MMStar, MathVerse, WeMath, LogicVista, MMVet) and across different model families (Qwen2.5-VL-3B/7B, InternVL3-2B) and RL algorithms (GRPO, RLOO), consistently yielding comparable or better pass@1 than full-data training and several strong selection baselines, while using a small fraction of the data and reducing wall-clock cost.  The ablations and sensitivity analyses around CDE, ACE, and DRM help demystify how each component contributes, and the dynamic-selection discussion clarifies why the authors ultimately choose a static variant. Overall, RAP is a clear, well-executed, and practically useful contribution for improving the efficiency and effectiveness of multimodal RL post-training.

**Weaknesses:**

1.Static, base-model-conditioned selection. RAP performs a one-shot selection of “cognitive samples” using the pre-RL base model and then keeps this subset fixed for all subsequent RL updates.  This inevitably ties the selected data distribution to the initial model’s inductive biases and failure modes: if the base model systematically under-utilizes certain reasoning patterns, those samples may be underrepresented or discarded. The paper does provide evidence that RAP generalizes across backbones and RL algorithms, and briefly discusses a dynamic re-selection variant, but there is no systematic study of when static selection becomes brittle (e.g., under much weaker or differently trained base models) or of lightweight adaptive schemes. This limits our understanding of RAP’s robustness beyond the specific settings tested.

2.Heuristic nature and uniqueness of the core metrics. The three components—CDE discrepancy, ACE attention confidence, and DRM difficulty—are instantiated via relatively simple empirical rules: 0/1 correctness gaps with global thresholding for CDE, a multiplicative attention score with a fixed λₐ for ACE, and group-wise accuracy with top-k replacement for DRM.  Although these choices are motivated and ablated, the paper does not compare them against alternative proxies (e.g., reward gaps, attention entropy/top-k mass, different difficulty statistics), nor provide a stronger argument that they are particularly well aligned with “reasoning activation potential.” As a result, it remains unclear whether RAP’s gains rely on these specific formulations or would largely carry over to simpler or more theoretically grounded metrics.

**Questions:**

1.On static vs. dynamic selection and robustness. How sensitive is the set of selected “cognitive samples” to the choice of base model and its initialization? For example, have you tried running RAP with a significantly weaker or differently fine-tuned base model and comparing (i) the overlap of selected samples and (ii) downstream RL performance? Relatedly, could you provide more concrete details or results on your dynamic re-selection experiments (Section 4.3 / Appendix A.2.2)—e.g., what re-selection schedule you tried, whether partial or online refreshing (rather than full re-scoring) was considered, and how these design choices influenced both compute cost and stability?

2.On metric design and possible alternatives. Have you experimented with alternative or simpler proxies for any of CDE, ACE, or DRM—for instance, using reward gaps instead of 0–1 correctness differences for discrepancy, or attention entropy / top-k mass instead of the multiplicative ψ(A), or different difficulty metrics beyond group-wise accuracy? If so, how did they compare? More broadly, how sensitive are your results to λ_c, λ_a, and top-k, and do you expect these hyperparameters to transfer across models and datasets without extensive retuning? Clarifying whether you view the current CDE/ACE/DRM formulation as essential to RAP, or as one effective instantiation in a broader family of possible metrics, would strengthen the paper.

**Details Of Ethics Concerns:**

No concern

---

### Note · Authors · 2025-12-22

I have read and agree with the venue's withdrawal policy on behalf of myself and my co-authors.